# InfGraND: An Influence-Guided GNN-to-MLP Knowledge Distillation

**Amir Eskandari**                                            *amir.eskandari@queensu.ca*
*School of Computing*
*Queen's University*

**Aman Anand**                                                *aman.anand@queensu.ca*
*School of Computing*
*Queen's University*

**Elyas Rashno**                                              *elyas.rashno@queensu.ca*
*School of Computing*
*Queen's University*

**Farhana Zulkernine**                                 *farhana.zulkernine@queensu.ca*
*School of Computing*
*Queen's University*

**Reviewed on OpenReview:** *https://openreview.net/forum?id=lfzHR3YwlD*

## Abstract

Graph Neural Networks (GNNs) are the go-to model for graph data analysis. However, GNNs rely on two key operations—`aggregation` and `update`, which can pose challenges for low-latency inference tasks or resource-constrained scenarios. Simple Multi-Layer Perceptrons (MLPs) offer a computationally efficient alternative. Yet, training an MLP in a supervised setting often leads to suboptimal performance. Knowledge Distillation (KD) from a GNN teacher to an MLP student has emerged to bridge this gap. However, most KD methods either transfer knowledge uniformly across all nodes or rely on graph-agnostic indicators such as prediction uncertainty. We argue this overlooks a more fundamental, graph-centric inquiry: *"How important is a node to the structure of the graph?"* We introduce a framework, InfGraND, an **Inf**luence-guided **Gra**ph K**N**owledge **D**istillation from GNN to MLP that addresses this by identifying and prioritizing structurally influential nodes to guide the distillation process, ensuring that the MLP learns from the most critical parts of the graph. Additionally, InfGraND embeds structural awareness in MLPs through one-time multi-hop neighborhood feature *pre-computation*, which enriches the student MLP's input and thus avoids inference-time overhead. Our rigorous evaluation in transductive and inductive settings across seven homophilic graph benchmark datasets shows InfGraND consistently outperforms prior GNN to MLP KD methods, demonstrating its practicality for numerous latency-critical applications in real-world settings.

**Keywords:** Graph Neural Networks, Knowledge Distillation, GNN-to-MLP Knowledge Distillation

## 1   Introduction

Graph-structured data has become increasingly important in a range of applications. For instance, social networks use graphs to model user interactions to facilitate effective identification of misinformation within communities (Han et al., 2020; Fan et al., 2020; Sharma et al., 2024). Similarly, e-commerce websites employ graphs to capture user-product relationships to provide personalized recommendations that increase

revenue (Wu et al., 2022; Wang et al., 2021; Gao et al., 2023). More recently, graphs have been widely adopted as a powerful source of external knowledge for Large Language Models (LLMs) (Zhao et al., 2023) through Retrieval-Augmented Generation (RAG) systems, which enable both enhanced factual grounding and deeper contextual reasoning (Peng et al., 2024; Han et al., 2024). The widespread use of such graph-based applications demands efficient analytical methods.

Graph Neural Networks (GNNs) (Kipf & Welling, 2016; Hamilton et al., 2017; Veličković et al., 2017; Wu et al., 2023a; Liu et al., 2020; Wu et al., 2020; Zhou et al., 2020; Li et al., 2019; Chen et al., 2020) emerged as a powerful framework to process graph data. At their core, GNNs utilize a layer-by-layer message-passing mechanism to learn rich node-level representations, which has proven highly effective in the aforementioned applications. However, this academic success has not been fully translated into industrial practice. The high computational and memory demands of message-passing create significant bottlenecks during training and inference, limiting their use in production environments (Zhang et al., 2020; Min et al., 2021; Jia et al., 2020). To overcome these constraints, industry often relies on a much simpler alternative, the Multi-Layer Perceptron (MLP). Although resource efficient, MLPs show less competitive performance because they operate exclusively on node features, ignoring valuable structural knowledge within the graph (Zhang et al., 2022).

To address the performance-efficiency trade-off, one line of research focuses on developing more efficient GNN architectures (Bojchevski et al., 2019; Huang et al., 2018; Ying et al., 2018; Chen et al., 2018). A more recent and popular research direction leverages the efficiency of MLPs while preserving the expressive power of GNNs through Knowledge Distillation (KD). In KD, a well-trained GNN teacher model transfers its knowledge to an MLP student (Yang et al., 2021; Zhang et al., 2022; Gou et al., 2021; Hinton, 2015). The GNN-to-MLP distillation field has evolved along several directions. A significant body of work focuses on enhancing the student side. Some approaches increase the complexity of the student MLP by employing advanced architectures such as ensemble methods or Mixture-of-Experts (Lu et al., 2024; Rumiantsev & Coates, 2024). Others aim to enrich the MLP's input by injecting structural knowledge through positional encoders or structure-aware tokenizer (Tian et al., 2022; Chen et al., 2024; Yang et al., 2024). Although often effective, these strategies increase computational overhead. Moreover, they treat all graph nodes uniformly during distillation, overlooking the varying importance of different nodes.

This equal treatment has led to a paradigm of discriminative distillation (Wu et al., 2023c; 2024). These works use entropy to rank nodes and sample them accordingly, prioritizing nodes where the teacher is less confident. However, we argue that this approach is fundamentally graph-agnostic: it relies on the teacher GNN's confidence in its predicted label for each node rather than the node's structural role within the graph. In other words, they discriminate between nodes based on *How certain is the teacher GNN about this node's label?* This raises a fundamental question: should we not instead ask *How influential is this node within the structure of the graph?*

To answer this question, we propose InfGraND, an *influence-guided* knowledge distillation method built on a graph-aware influence metric that moves beyond prediction uncertainty. The influence score is a topology-aware indicator that measures how perturbations in a node's features affect the representations of the other nodes after message propagation. For the influence score, we leverage an influence maximization strategy inspired by previous work in the active learning literature (Li et al., 2018). Our preliminary experiments confirm that prioritizing high-influence nodes consistently yields superior teacher GNN performance (see empirical validation in Section 5.2.1, including Figure 2). Based on this observation, InfGraND employs a deterministic soft-weighting scheme in the distillation via a subgraph-level distillation loss (Yang et al., 2020). It discriminates among neighbors in the subgraph based on their influence score. This influence metric is parameter-free. To further incorporate structural knowledge at the input level, InfGraND enriches the input features of the student. Inspired by practices in large-scale industrial systems (Li et al., 2013) like pre-computed embedding tables, we utilize an efficient one-time feature propagation and pooling operation. This approach allows the MLP to access rich multi-hop neighborhood information without adding inference overhead.

Our evaluation covers both transductive (i.e., training and testing on the same graph) and inductive (i.e., training on one graph and testing on another) settings. Widely adopted GNN teachers such as Graph

Convolutional Network (GCN) (Kipf & Welling, 2016), Graph Attention Network (GAT) (Veličković et al., 2017), and GraphSAGE (Hamilton et al., 2017) are used, with MLPs serving as students. We benchmark InfGraND against state-of-the-art (SOTA) GNN-to-MLP models, particularly those designed for non-uniform and discriminative distillation. We also conduct experiments in scenarios where labels are limited, as well as comprehensive ablation and sensitivity analyses to provide further insights into the behavior of the model. The following are our main contributions.

- We provide a taxonomy of GNN-to-MLP distillation methods, revealing that existing approaches either add computational complexity or employ graph-agnostic discrimination strategies.
- We propose InfGraND, a novel framework that, to the best of our knowledge, is the first to perform node-level discrimination by computing node influence based on the graph structure in GNN-to-MLP distillation.
- We validate our framework through rigorous evaluation on seven real-world homophilic graph benchmark datasets in both transductive and inductive settings. Our experimental results demonstrate that InfGraND consistently outperforms these competing approaches. It not only substantially outperforms vanilla MLPs and existing state-of-the-art distillation methods, but in many cases, even surpasses the performance of its own GNN teachers. Results from label-limited scenarios and ablation studies further validate the effectiveness of our proposed method.

The remainder of the paper is structured as follows. In Section 2, we present a review of relevant research and categorize the GNN-to-MLP distillation paradigm. Section 3 provides the necessary background concepts to support our approach. Our proposed method and its components are detailed in Section 4. Section 5 outlines the experimental setup and presents the results. We conclude in Section 6 with a summary of our contributions and a discussion of future research directions.

## 2 Related Work

The evolution of GNNs reflects ongoing efforts to balance expressiveness with computational efficiency. The early spectral methods (Bruna et al., 2013; Henaff et al., 2015; Defferrard et al., 2016; Kipf & Welling, 2016) leveraged graph Fourier transforms but faced scalability challenges. Spatial GNNs(Micheli, 2009; Scarselli et al., 2008; Hamilton et al., 2017; Veličković et al., 2017) improved scalability through localized message passing, yet the recursive nature of aggregation still limits deployment at scale. GNN-to-MLP distillation has emerged as a more recent approach to eliminate message passing while preserving the knowledge learned by GNN teachers.

### 2.1 Distilling Graph Knowledge into MLPs

Graph-less Neural Networks (GLNN) (Zhang et al., 2022) established the foundational framework by training a student MLP on the soft labels from a GNN teacher. Since then, the field has evolved along several directions. One line of work increases the student MLP's capacity to improve performance. For instance, AdaGMLP (Lu et al., 2024) uses an AdaBoost-style ensemble of MLPs, while RbM (Rumiantsev & Coates, 2024) proposes a Mixture-of-Experts (MoE) student model that enforces expert specialization on different regions of the representation space. However, these methods increase the model complexity and inference overhead. Another research direction enriches the input features of MLPs with explicit structural knowledge. NOSMOG (Tian et al., 2022) incorporates positional features from DeepWalk (Perozzi et al., 2014), concatenating them with node features to make the student MLP structure-aware. SA-MLP (Chen et al., 2024) directly encodes the adjacency matrix with a linear layer to integrate structural knowledge. VQGraph (Yang et al., 2024) learns a structure-aware tokenizer to create a discrete codebook of local graph structures for more expressive distillation targets. These strategies add computational overhead by expanding the input dimension of the student MLP and often require separate training for positional information. Despite these advances, both lines of work treat all nodes uniformly during distillation, overlooking that nodes vary in their importance within the graph structure.

**Non-Uniform Distillation.** A third line of work discriminates between nodes using prediction uncertainty. KRD (Wu et al., 2023c) quantifies "knowledge reliability" via prediction entropy stability under noise and

samples more reliable nodes during training. HGMD (Wu et al., 2024) defines "knowledge hardness" via entropy and extracts hardness-aware subgraphs as additional supervision for challenging samples. However, entropy-based metrics are graph-agnostic, assessing prediction confidence rather than structural influence.

We address this gap by introducing a graph-aware node importance metric that quantifies structural influence rather than prediction confidence. We also incorporate structural features efficiently via one-time pre-computation, eliminating the inference overhead of existing input augmentation approaches.

## 3 Background

**Notations.** Consider $\mathcal{G} = (\mathcal{V}, \mathcal{E}, \mathbf{X})$ as an attributed graph, where $\mathcal{V}$ is the set of $N$ nodes with features $\mathbf{X} = [\mathbf{x}_1, \mathbf{x}_2, \cdots, \mathbf{x}_N] \in \mathbb{R}^{N \times d}$ and $\mathcal{E}$ denotes the edge set. A $d$-dimensional features vector $\mathbf{x}_i$ is assigned for each node $v_i \in \mathcal{V}$. Each edge $e_{i,j} \in \mathcal{E}$ denotes a connection between nodes $v_i$ and $v_j$. The graph structure is represented by an adjacency matrix $\mathbf{A}^{N \times N} \in [0, 1]$ with $\mathbf{A}_{i,j} = 1$ if $e_{i,j} \in \mathcal{E}$ and $\mathbf{A}_{i,j} = 0$ if $e_{i,j} \notin \mathcal{E}$. Also for each node, we have an assigned label $\mathbf{y}_i \in \{0, 1, \ldots, C-1\}$ where $C$ is the number of classes.

**Node Classification.** In semi-supervised node classification tasks, only a subset of nodes $\mathcal{V}_{\mathrm{lab}}$ with labels $\mathbf{Y}_{\mathrm{lab}}$ are known. The $\mathcal{V}_{\mathrm{lab}}$ nodes are labeled as the set $\mathcal{D}_{\mathrm{lab}} = (\mathcal{V}_{\mathrm{lab}}, \mathbf{Y}_{\mathrm{lab}})$. The unlabeled set is defined as $\mathcal{D}_{\mathrm{unl}} = (\mathcal{V}_{\mathrm{unl}}, \mathbf{Y}_{\mathrm{unl}})$, where $\mathcal{V}_{\mathrm{unl}} = \mathcal{V} \setminus \mathcal{V}_{\mathrm{lab}}$. The node classification task aims to learn a mapping $\Phi : \mathbf{X} \to \mathbf{Y}$ via $\mathbf{Y}_{\mathrm{lab}}$ to infer $\mathbf{Y}_{\mathrm{unl}}$. Each label is typically represented as a one-hot vector in $\mathbf{Y} \in \mathbb{R}^{N \times C}$. The transductive setting utilizes the full graph $\mathcal{G}_{\mathrm{train}} = \mathcal{G} = (\mathcal{V}, \mathcal{E}, \mathbf{X})$ where $\mathcal{V} = \mathcal{V}_{\mathrm{lab}} \cup \mathcal{V}_{\mathrm{unl}}$ during training, with access to all node features $\mathbf{X} = [\mathbf{x}_1, \mathbf{x}_2, ..., \mathbf{x}_N] \in \mathbb{R}^{N \times d}, \forall v_i \in \mathcal{V}$. The model is then evaluated on the nodes $\mathcal{V}_{\mathrm{unl}}$ encountered during training. Under the inductive setting, the model is trained on an observed subgraph $\mathcal{G}_{\mathrm{obs}} = (\mathcal{V}_{\mathrm{obs}}, \mathcal{E}_{\mathrm{obs}}, \mathbf{X}_{\mathrm{obs}})$, where $\mathcal{E}_{\mathrm{obs}} = \{e_{i,j} \in \mathcal{E} \mid v_i, v_j \in \mathcal{V}_{\mathrm{obs}}\}$ and $\mathcal{V}_{\mathrm{lab}} \subseteq \mathcal{V}_{\mathrm{obs}}$. The model is then tested on completely unseen nodes and their edges in $\mathcal{G}_{\mathrm{unobs}}$, where $\mathcal{V}_{\mathrm{test}} \subseteq \mathcal{V}_{\mathrm{unobs}}$, to evaluate its ability to generalize to new graph structures.

**Graph Neural Networks (GNNs).** Most existing GNNs follow a message-passing scheme that consists of two key computations for each node $v_i$: (1) AGGREGATE: aggregates messages from neighborhood $\mathcal{N}(v_i)$; (2) UPDATE: updates node representation based on the output of the previous layer and aggregated messages. For a $L$-layer GNN, the formulation of the $l$-th layer is the following:

$$\mathbf{h}_i^{(l)} = \mathrm{UPDATE}^{(l)} \left( \mathbf{h}_i^{(l-1)}, \mathbf{m}_i^{(l)} \right), \qquad \mathbf{m}_i^{(l)} = \mathrm{AGGREGATE}^{(l)} \left( \{ \mathbf{h}_j^{(l-1)} : v_j \in \mathcal{N}(v_i) \} \right), \tag{1}$$

where $1 \leq l \leq L$, $\mathbf{h}_i^{(l)}$ is the representation of node $v_i$ at the $l$-th layer, and $\mathbf{m}_i^{(l)}$ is the aggregated message from its neighbors. The process is initialized with the input features, where $\mathbf{h}_i^{(0)} = \mathbf{x}_i$. Common GNN variants include GCN (Kipf & Welling, 2016), GraphSAGE (Hamilton et al., 2017), and GAT (Veličković et al., 2017).

## 4 Proposed Method

This section describes the proposed InfGraND framework in detail. First, we introduce our node influence measurement to quantify node importance (Section 4.1). Next, Section 4.2 explains the *pre-computation* step that enables the MLP to capture structural graph knowledge. The section concludes by presenting the full influence-guided distillation process (Section 4.3).

### 4.1 Node Influence Measurement

To determine node importance, we adopt a graph-aware node influence framework that quantifies how perturbations to a single node's features propagate through the graph to affect the representations of other nodes (Zhang et al., 2021). Therefore, a node with a greater effect on the graph is considered topologically more influential.

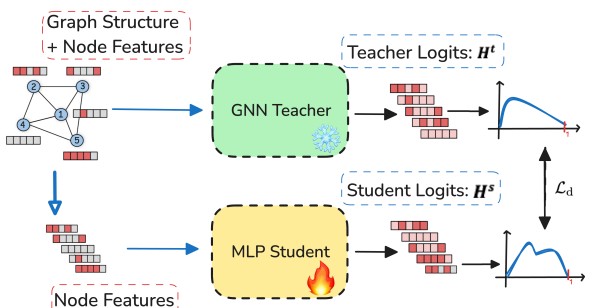
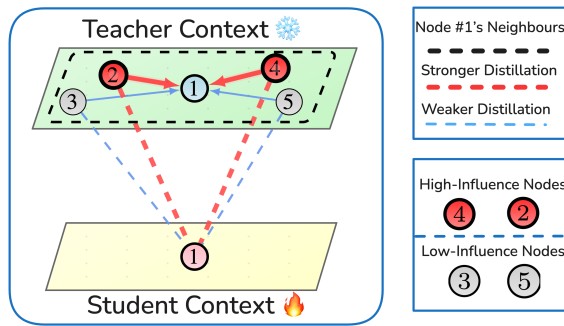

(a) The GNN Teacher, trained using supervised learning, remains frozen during distillation. The MLP Student, which only sees node features, learns structural information by mimicking the teacher output.

(b) In subgraph-level KD, the student learns the representation of node 1 from neighbors provided by the teacher, weighted by influence scores. High-influence neighbors yield stronger supervision.

Figure 1: **Overview of Influence-Guided Distillation.** Our method uses node structural influence to weight the knowledge transfer, ensuring that the student learns local structure from its most important neighbors.

**Definition 4.1 (Quantifying Node Influence).** Within a graph $\mathcal{G} = (\mathcal{V}, \mathcal{E}, \mathbf{X})$, we define the node influence score of a source node $v_i$ on a target node $v_j$ after $k$ message-passing iterations as the L1-norm of the expected Jacobian matrix:

$$\hat{I}_{(j \leftarrow i)}(v_j, v_i, k) = \left\| \mathbb{E}\left[\frac{\partial \mathbf{x}_j^{(k)}}{\partial \mathbf{x}_i^{(0)}}\right] \right\|_1, \tag{2}$$

where $\mathbf{x}_j^{(k)}$ represents the feature embedding of the target node $v_j$ at the $k$-th iteration, and $\mathbf{x}_i^{(0)}$ is the initial features of the source node $v_i$. To provide a relative measure of influence, we use a normalized influence score:

$$I_{(j \leftarrow i)}(v_j, v_i, k) = \frac{\hat{I}_{(j \leftarrow i)}(v_j, v_i, k)}{\sum_{v_w \in \mathcal{V}} \hat{I}_{(j \leftarrow w)}(v_j, v_w, k)}. \tag{3}$$

This normalized score, $I_{(j \leftarrow i)}(v_j, v_i, k)$, represents the proportion of influence of $v_i$ on $v_j$ relative to the total influence of all other nodes that feed to $v_j$ in the graph.

To practically measure this influence, we must approximate the expected Jacobian term in Eq. 2. A direct calculation is often intractable. Previous work establishes that the expected influence is equivalent to the aggregated influence on all $k$-length random walks between two nodes (Xu et al., 2018). Inspired by Simplifying Graph Convolutional Networks (SGC) (Wu et al., 2019), we remove the non-linear activations and weight matrices. This simplifies the GCN down to its core function of pure topological propagation, defined as:

$$\mathbf{X}^{(k)} = \tilde{\mathbf{A}} \mathbf{X}^{(k-1)}, \tag{4}$$

where $\tilde{\mathbf{A}}$ is the normalized adjacency matrix, defined as $\tilde{\mathbf{A}} = \tilde{\mathbf{D}}^{-\frac{1}{2}}(\mathbf{A} + \mathbf{I})\tilde{\mathbf{D}}^{-\frac{1}{2}}$, with $\tilde{\mathbf{D}}$ being the degree matrix. We use $k = 2$ (see Appendix H for sensitivity analysis) so that the resulting embedding $\mathbf{x}_j^{(2)}$ contains information from its 2-hop neighborhood. We therefore use the cosine similarity, $\text{sim}_{\cos}(\mathbf{x}_i^{(0)}, \mathbf{x}_j^{(2)})$, as an efficient and parameter-free indicator to determine the influence of node $v_i$ on $v_j$. To ensure that the resulting influence scores lie between 0 and 1, we apply MinMaxScaler (Komer et al., 2014) globally per graph. Influence computation is a *one-time* preprocessing step during training, not a recurring cost during inference. More details about the influence computation are provided in Appendix F.

To enhance knowledge distillation using a node influence score, instead of a pairwise score (Eq. 3), we define a ***Global Influence Score*** (GIS) that measures the overall impact of each node on the entire graph and assign that to each node.

**Definition 4.2** (**Global Influence Score**). The global influence score $\mathcal{I}_g(v_i)$ of node $v_i$ after $k$ message passing iteration is defined as:

$$\mathcal{I}_g(v_i) = \frac{\sum_{j \in \mathcal{V}} I_{(j \leftarrow i)}(v_j, v_i, k)}{\max\limits_{l \in \mathcal{V}} \left( \sum_{j \in \mathcal{V}} I_{(j \leftarrow l)}(v_j, v_l, k) \right)}, \tag{5}$$

where $I_{(j \leftarrow i)}(v_j, v_i, k)$ is defined in Eq. 3. In Eq. 5, the numerator is normalized by the maximum global influence score across all nodes, which ensures that $\mathcal{I}_g(v_i)$ lies within the range $[0, 1]$.

## 4.2 Node Feature Propagation

To provide the student MLP with structural knowledge, InfGraND incorporates an efficient feature propagation scheme. We perform this as a *one-time, offline pre-computation* step before training begins. This approach is inspired by common practices in large-scale industrial systems such as using a parameter server to manage precomputed embedding tables (Li et al., 2013). We use the linear propagation from Eq. 4 to generate multi-hop feature matrices, $\{\mathbf{X}^{(p)}\}_{p=0}^{P}$. To avoid increasing the input dimensionality or adding parameters, we apply average pooling across these matrices instead of concatenation:

$$\tilde{\mathbf{X}} = \text{POOL}\left( \{\mathbf{X}^{(p)}\}_{p=0}^{P} \right). \tag{6}$$

The resulting matrix, $\tilde{\mathbf{X}}$, contains multi-hop neighborhood information. It serves as the fixed input to the student MLP. This pooling is computed once during an offline propagation step. As a result, there is no added cost at inference time. The model remains efficient during deployment.

## 4.3 Distillation

An overview of the distillation mechanism is provided in Figure 1 (a). The training process starts with standard supervised training of the teacher GNN. Once trained, the teacher is frozen. The student MLP is then trained using a composite objective that learns from both the ground-truth labels and the soft predictions provided by the teacher.

To ground the student in the true class distributions, we use an *influence-weighted supervised loss*, $\mathcal{L}_s$. This loss is applied only to the labeled nodes, $\mathcal{V}_{\text{lab}}$:

$$\mathcal{L}_s = \delta_1 \sum_{v_i \in \mathcal{V}_{\text{lab}}} D_{\text{CE}}(\sigma(\mathbf{h}_i^s), \mathbf{y}_i) + \delta_2 \sum_{v_i \in \mathcal{V}_{\text{lab}}} \mathcal{I}_g(v_i) \cdot D_{\text{CE}}(\sigma(\mathbf{h}_i^s), \mathbf{y}_i), \tag{7}$$

where $D_{\text{CE}}$ denotes the standard cross-entropy loss, $\mathbf{h}_i^s$ is the student's representation for node $v_i$, $\sigma(\cdot)$ is the softmax function, and $\mathcal{I}_g(v_i)$ is the global influence score of node $v_i$. The hyperparameters $\delta_1$ and $\delta_2$ control the contribution of the standard and influence-weighted loss terms, respectively.

For KD, our *primary loss*, $\mathcal{L}_d$, leverages the homophily principle common in graphs (Yang et al., 2020). Training encourages the student's prediction for node $i$, $\mathbf{h}_i^s$, to be similar to the teacher's predictions for its neighboring nodes $j$, $\mathbf{h}_j^t$, via a Kullback–Leibler (KL) divergence loss, denoted as $D_{\text{KL}}$. The term $\tau$ denotes the distillation temperature. As illustrated in Figure 1 (b), the influence score of the teacher's node, $\mathcal{I}_g(v_j)$, directly weights the distillation process, allowing high-influence neighbors to provide a stronger distillation signal. The loss is defined as:

$$\mathcal{L}_d = \sum_{i \in \mathcal{V}} \sum_{j \in \mathcal{N}(v_i)} (\gamma_1 + \gamma_2 \cdot \mathcal{I}_g(v_j)) \cdot \frac{1}{|\mathcal{N}(v_i)|} \cdot D_{\text{KL}}(\sigma(\mathbf{h}_i^s / \tau) \parallel \sigma(\mathbf{h}_j^t / \tau)). \tag{8}$$

The design of this loss is crucial. The $\gamma_1$ term provides a baseline distillation gradient from all neighbors, while the $\gamma_2 \cdot \mathcal{I}_g(v_j)$ term acts as a fine-grained amplifier for more influential nodes. We provide a full

theoretical analysis of this loss function, including a derivation of its gradients (Appendix D.1), the effect of the influence score (Appendix D.2), and the need for $\gamma_1$ (Appendix D.3) to rigorously justify this formulation.

The overall training objective for the student model combines these two losses:

$$\mathcal{L}_{\text{t}} = \lambda\mathcal{L}_{\text{s}} + (1 - \lambda)\mathcal{L}_{\text{d}}, \tag{9}$$

where $\lambda \in [0, 1]$ is a hyperparameter that balances the supervised and distillation signals. The complete training procedure is presented in Algorithm 1 (Appendix G).

## 5 Experiments and Results

We conduct rigorous experiments to comprehensively evaluate the effectiveness of our InfGraND framework by addressing six key research questions. **Q1:** Does training a GNN on high-influence nodes lead to better performance than using low-influence nodes? **Q2:** How does InfGraND perform on node classification tasks compared to baseline GNN-to-MLP distillation methods, its corresponding GNN teacher models trained with supervised loss, and a vanilla MLP trained without distillation? **Q3:** What is the trade-off between classification accuracy and inference latency for InfGraND compared to alternatives? **Q4:** What is the relative contribution of each component of InfGraND to its final performance? **Q5:** How robust is InfGraND's performance when the number of available training labels is severely limited? **Q6:** What is the impact of key hyperparameters on the performance of InfGraND, specifically the influence-related loss weights $(\gamma_2, \delta_2)$, the number of propagation steps $(P)$, and the choice of pooling method?

### 5.1 Experimental Setting

**Datasets.** We evaluate InfGraND in both transductive and inductive settings on seven real-world datasets with inherent graph structures: (1) Cora (Sen et al., 2008), (2) Citeseer (Giles et al., 1998), (3) Pubmed (McCallum et al., 2000), (4) Amazon-Photo, (5) CoAuthor-CS, (6) CoAuthor-Phy (Shchur et al., 2018), and (7) the large-scale OGBN-Arxiv dataset (Hu et al., 2020). For small-scale citation datasets (Cora, Citeseer, and Pubmed), we use the splitting strategy of Kipf et al. (2016). For CoAuthor-CS, CoAuthor-Phy, and Amazon-Photo, we adopt the random split method as used by Yang et al. (2021) and Zhang et al. (2022). Finally, for the OGBN-Arxiv dataset, we use the official splits from Hu et al. (2020). We choose a random seed and apply it consistently to ensure identical splits across experiments for fair and reproducible evaluation; different seeds produce different splits. Dataset details and splitting statistics are discussed in the Appendix A.

**Implementation.** We utilize three GNN architectures as teachers: GCN (Kipf & Welling, 2016), GAT (Veličković et al., 2017), and GraphSAGE (Hamilton et al., 2017). We select these models as they represent diverse and widely-adopted design paradigms. We benchmark InfGraND against a strong set of competitive baselines, including the foundational GLNN (Zhang et al., 2022) and the non-uniform distillation frameworks KRD (Wu et al., 2023c), HGMD (Wu et al., 2024), and FF-G2M (Wu et al., 2023b). We reproduced the results for all baselines using their official public implementations. This step was crucial because our experimental settings and inductive data splits for certain datasets and teacher models differed from those in the original papers. Note that at the time of our experiments, the official HGMD implementation only supported the transductive setting with a GCN teacher, limiting our comparison to that setup. To ensure reproducibility, we used a fixed set of five different random seeds and reported the average performance on five runs. The hyperparameters were tuned using the WandB platform (Biewald, 2020), with validation accuracy as the tuning criterion. To ensure every model was evaluated at its peak, we performed random hyperparameter searches for all methods, including baselines and our proposed InfGraND, until performance on the validation set saturated. Appendix B provides additional information on reproducibility. Our implementation [1] uses PyTorch (Paszke et al., 2019) and the DGL library (Wang et al., 2019), with experiments run on a server equipped with an NVIDIA V100 GPU (32GB VRAM).

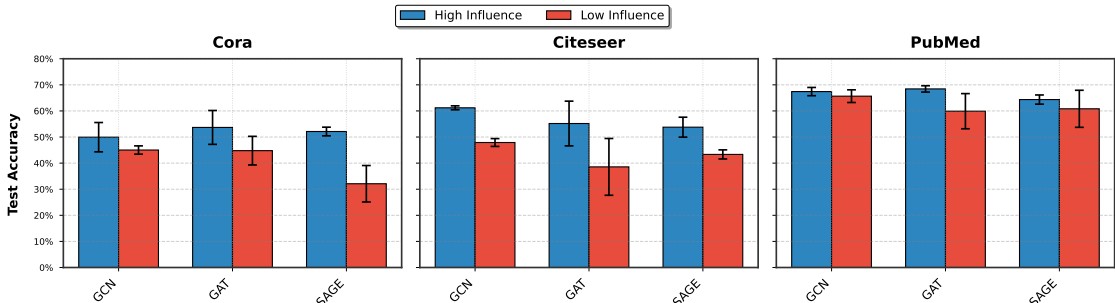

Figure 2: Test set classification accuracy of three GNN models (GCN, GAT, GraphSAGE) when trained on a subset of high-influence vs. low-influence nodes across different datasets (Cora, Citeseer, PubMed). Error bars represent standard deviation over five runs.

## 5.2 Evaluation

### 5.2.1 Q1 - Effect of Influence on GNNs

We evaluate the impact of training GNNs on nodes with different influence scores in the transductive setting. To do so, we divide the training set into two subsets, each containing 25% of all labeled nodes. We ensure the selection preserves class balance. One subset contains high-influence nodes (top 25% per class), the other low-influence nodes (bottom 25%). Then, we train separate GNNs (GCN, GAT, GraphSAGE) on each subset using a fixed test set.

**Discussion.** As shown in Figure 2, models trained on high-influence nodes (blue bars) consistently outperform those trained on low-influence nodes (red bars) across all datasets (Cora, Citeseer, and PubMed). This consistent performance gap provides strong empirical evidence that the influence score effectively identifies nodes most critical to a GNN's generalization. This finding directly motivates the design of InfGraND, which prioritizes knowledge transfer from high-influence nodes during the distillation process.

### 5.2.2 Q2 - Classification Performance

Since we observed that training a teacher on high-influence nodes improves generalization, we prioritize these nodes during distillation, which should enhance the performance of the distilled MLP. Empirical evidence supporting this hypothesis is provided in Table 1 and Figure 3. Table 1 reports node classification results in the transductive and inductive settings. Figure 3 presents results on the large-scale OGBN-Arxiv dataset for InfGraND, KRD, GraphSAGE, and Vanilla MLP.

**Discussion.** All the results support the hypothesis and demonstrate the following: (1) InfGraND on average outperforms all baselines in both transductive and inductive settings and achieves the highest accuracy in most cases; (2) in nearly all cases, the distilled MLPs surpass their GNN teachers, a key finding that challenges the assumption that expressive GNNs are always superior to MLPs on graph data; (3) the performance gains over non-distilled MLPs are substantial. As shown in Table 1, InfGraND effectively bridges the gap between MLPs and GNNs. Compared to the vanilla MLP, the InfGraND-distilled MLP achieves an average improvement of 12.6% under the transductive setting and 9.3% under the inductive setting, which corresponds to the average $\Delta$ across all three teacher architectures; (4) InfGraND outperforms discriminative distillation methods, including hardness- and reliability-based approaches, with average gains of 0.9% over KRD (transductive), 3.0% (inductive), and 0.6% over HGMD, based on the $\Delta$ values in Table 1; (5) InfGraND scales well to large graphs, outperforming the KRD baseline on OGBN-Arxiv under both transductive and inductive settings (Figure 3); (6) a stronger teacher does not necessarily yield a stronger student. On Amazon-Photo (transductive), GCN outperforms GAT (90.7% vs. 87.6%), yet the GAT-distilled student achieves higher accuracy (94.5% vs. 94.2%), indicating that teacher accuracy alone does not determine distillation effectiveness.

---

[1] https://github.com/AmEskandari/InfGraND

Table 1: Node classification accuracy (%) for various models under transductive and inductive settings, averaged over 5 runs. Boldface indicates best performance. Dataset abbreviations: Amazon = Amazon-Photo, CS = Coauthor-CS, Phy = Coauthor-Physics. Gray-shaded rows correspond to models trained using only supervised loss functions (i.e., without distillation). $\Delta$ denotes the average accuracy improvement of InfGraND over the corresponding baseline method across all datasets.

| Teacher | Method | Transductive Accuracy | | | | | | | Inductive Accuracy | | | | | | |
|---|---|---|---|---|---|---|---|---|---|---|---|---|---|---|---|
| | | Cora | Citeseer | Pubmed | Amazon | CS | Phy | $\Delta$ | Cora | Citeseer | Pubmed | Amazon | CS | Phy | $\Delta$ |
| GCN | Vanilla GCN | 82.2 ± 0.6 | 71.6 ± 0.2 | 79.2 ± 0.3 | 90.7 ± 0.3 | 89.3 ± 0.0 | 91.9 ± 1.3 | +3.0 | 80.6 ± 1.4 | 64.8 ± 0.1 | 71.6 ± 2.2 | 88.8 ± 1.2 | 89.4 ± 0.5 | 90.0 ± 1.3 | +2.5 |
| | Vanilla MLP | 57.8 ± 1.0 | 60.5 ± 0.7 | 72.8 ± 0.4 | 79.0 ± 1.0 | 87.8 ± 0.5 | 89.5 ± 2.0 | +12.6 | 58.4 ± 2.5 | 55.1 ± 1.0 | 70.8 ± 1.2 | 78.6 ± 1.2 | 88.1 ± 1.3 | 89.0 ± 0.9 | +10.0 |
| | GLNN | 83.1 ± 0.3 | 73.0 ± 0.5 | 79.4 ± 0.6 | 92.3 ± 0.5 | 92.6 ± 0.4 | 93.6 ± 1.1 | +1.5 | 71.0 ± 1.7 | 65.0 ± 1.5 | 72.5 ± 0.8 | 88.1 ± 1.8 | 88.6 ± 2.9 | 90.9 ± 2.5 | +4.0 |
| | KRD | 83.3 ± 0.9 | 73.9 ± 0.8 | **81.8** ± 0.4 | 91.7 ± 1.5 | 93.1 ± 0.5 | 94.1 ± 0.3 | +0.8 | 71.2 ± 0.4 | 65.0 ± 0.0 | **75.0** ± 0.3 | 87.3 ± 2.8 | 90.2 ± 1.9 | 91.6 ± 3.5 | +3.3 |
| | FF-G2M | 83.5 ± 0.7 | 74.0 ± 0.5 | 79.9 ± 0.4 | 93.0 ± 0.2 | 93.0 ± 0.5 | 93.7 ± 1.5 | +1.0 | 71.1 ± 0.6 | 65.8 ± 2.0 | 72.8 ± 0.5 | 88.8 ± 2.1 | 89.2 ± 1.4 | 91.8 ± 3.0 | +3.5 |
| | HGMD-mixup | 83.9 ± 2.0 | 74.6 ± 0.1 | 81.9 ± 0.2 | 92.3 ± 1.3 | 93.1 ± 0.5 | 93.4 ± 1.3 | +0.6 | - | - | - | - | - | - | |
| | InfGraND | **84.0** ± 0.5 | **75.2** ± 1.1 | 81.3 ± 0.2 | **94.2** ± 0.4 | **93.5** ± 0.6 | **94.7** ± 0.0 | | **81.5** ± 0.3 | **68.4** ± 0.5 | **75.0** ± 0.6 | **90.7** ± 0.6 | **91.8** ± 0.7 | **92.9** ± 1.6 | |
| SAGE | Vanilla SAGE | 82.5 ± 0.6 | 70.8 ± 0.6 | 77.9 ± 0.4 | 92.6 ± 0.3 | 89.7 ± 0.0 | 92.0 ± 0.9 | +2.8 | 79.6 ± 1.5 | 64.7 ± 0.8 | 73.0 ± 2.0 | 91.2 ± 0.8 | 89.0 ± 0.7 | 90.5 ± 1.7 | +2.1 |
| | Vanilla MLP | 57.8 ± 1.0 | 60.5 ± 0.7 | 72.8 ± 0.4 | 79.0 ± 1.0 | 87.8 ± 0.5 | 89.5 ± 2.0 | +12.5 | 58.4 ± 2.5 | 55.1 ± 1.0 | 70.8 ± 1.2 | 78.6 ± 1.2 | 88.1 ± 1.3 | 89.0 ± 0.9 | +10.1 |
| | GLNN | 83.2 ± 0.9 | 70.4 ± 1.9 | 79.2 ± 0.5 | 92.4 ± 0.5 | 92.3 ± 1.0 | 93.6 ± 1.5 | +1.9 | 69.6 ± 1.7 | 64.0 ± 1.1 | 72.4 ± 0.5 | 85.0 ± 2.2 | 89.3 ± 0.7 | 91.0 ± 3.0 | +4.8 |
| | KRD | 83.6 ± 1.0 | 73.8 ± 0.6 | 80.9 ± 0.5 | 91.7 ± 1.3 | 93.2 ± 0.7 | 94.1 ± 1.0 | +0.9 | 71.4 ± 0.4 | 65.5 ± 0.0 | **75.0** ± 0.0 | 88.4 ± 2.3 | 91.2 ± 1.8 | 91.0 ± 3.0 | +3.0 |
| | FF-G2M | 83.9 ± 0.8 | 72.8 ± 0.6 | 79.5 ± 0.5 | 92.3 ± 0.7 | 92.8 ± 0.7 | 93.5 ± 1.5 | +1.3 | 69.9 ± 0.7 | 65.6 ± 1.7 | 73.5 ± 0.5 | 88.1 ± 1.8 | 90.1 ± 1.8 | 92.9 ± 1.3 | +3.4 |
| | InfGraND | **84.5** ± 0.6 | **74.3** ± 0.5 | **81.3** ± 0.4 | **94.6** ± 0.3 | **93.4** ± 0.5 | **94.5** ± 1.1 | | **79.9** ± 0.6 | **67.7** ± 1.1 | 74.3 ± 1.1 | **93.5** ± 1.6 | **91.8** ± 0.7 | **93.3** ± 3.0 | |
| GAT | Vanilla GAT | 81.8 ± 1.2 | 70.4 ± 0.9 | 77.5 ± 0.2 | 87.6 ± 1.6 | 90.5 ± 0.0 | 91.9 ± 1.2 | +3.8 | **80.1** ± 2.2 | 65.8 ± 1.6 | 71.9 ± 0.8 | 88.6 ± 1.6 | 90.0 ± 1.2 | 90.2 ± 5.1 | +1.9 |
| | Vanilla MLP | 57.8 ± 1.0 | 60.5 ± 0.7 | 72.8 ± 0.4 | 79.0 ± 1.0 | 87.8 ± 0.5 | 89.5 ± 2.0 | +12.6 | 58.4 ± 2.5 | 55.1 ± 1.0 | 70.8 ± 1.2 | 78.6 ± 1.2 | 88.1 ± 1.3 | 89.0 ± 0.9 | +9.7 |
| | GLNN | 83.4 ± 0.4 | 70.6 ± 2.5 | 80.5 ± 2.4 | 91.5 ± 0.6 | 93.3 ± 0.5 | 93.3 ± 1.4 | +1.7 | 70.3 ± 0.8 | 63.5 ± 1.6 | 72.3 ± 0.6 | 85.8 ± 2.2 | 89.8 ± 2.1 | 92.0 ± 2.3 | +3.8 |
| | KRD | 83.0 ± 1.1 | 72.9 ± 0.6 | 81.4 ± 0.4 | 91.8 ± 1.4 | **94.3** ± 0.5 | 94.0 ± 1.3 | +0.9 | 73.0 ± 0.0 | 66.0 ± 0.0 | 74.9 ± 0.6 | 87.6 ± 3.4 | 89.0 ± 4.0 | 91.0 ± 2.5 | +2.8 |
| | FF-G2M | 83.5 ± 0.6 | 71.4 ± 1.4 | 80.9 ± 0.6 | 91.0 ± 0.6 | 93.0 ± 0.3 | 94.0 ± 1.5 | +1.5 | 71.5 ± 1.8 | 63.2 ± 2.1 | 72.5 ± 1.2 | 89.3 ± 2.1 | 90.0 ± 2.2 | 92.0 ± 1.9 | +3.3 |
| | InfGraND | **84.2** ± 0.5 | **73.9** ± 0.8 | **81.6** ± 0.5 | **94.5** ± 0.3 | 94.2 ± 0.6 | **94.4** ± 0.0 | | 79.9 ± 0.5 | **67.3** ± 0.9 | **75.1** ± 0.7 | **91.8** ± 0.3 | **91.2** ± 1.8 | **93.0** ± 2.2 | |

### 5.2.3 Q3 - Computational Time and Efficiency

In Figure 4, we present the trade-off between accuracy and inference time under the transductive setting, using the Citeseer dataset as a representative example. To ensure fairness, we keep the number of layers and hidden dimensions consistent across MLP, InfGraND, AdaGMLP, and GLNN. While Tian et al. (2022) report that wider GLNNs with more hidden neurons achieve higher accuracy at the cost of longer inference time, in our experiments we did not observe this effect: simply increasing the number of hidden dimensions or layers did not consistently improve performance. In fact, even a small number of hidden dimensions was sufficient to reach 100% training accuracy, suggesting that greater model complexity does not necessarily yield better results. Therefore, given that additional complexity did not improve performance, we kept the hidden dimensions and layers of the distilled MLP methods fixed and did not report results for larger variants.

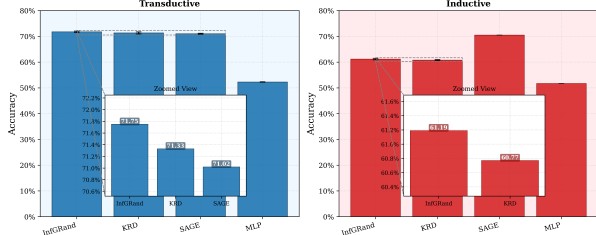

Figure 3: Transductive and inductive results on OGBN-Arxiv with a SAGE teacher. Zoomed-in views highlight the superior performance of InfGraND.

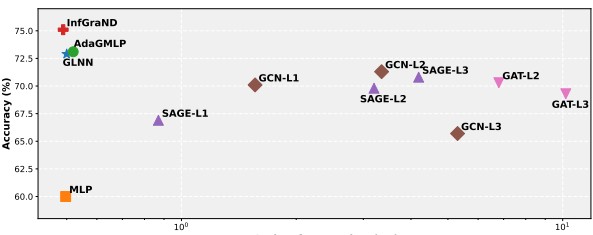

Figure 4: Trade-off between model accuracy and inference time on the Citeseer dataset. The x-axis shows inference time in milliseconds (log scale), and the y-axis shows classification accuracy.

**Discussion.** As shown in Figure 4, GraphSAGE, GCN, and GAT achieve the best results with 3, 2, and 2 layers, respectively. InfGraND gains 4.3% improvement over GraphSAGE-L3, 3.8% over GCN-L2, and 4.8% over GAT-L2 while being 8.56x, 6.84x, and 13.89x faster respectively. Also, AdaGMLP, which focuses on enhancing MLPs, required more time than InfGraND and GLNN, as it relies on an ensemble of models for prediction. We do not report results for FF-G2M, KRD, and HGMD, as they share the same architecture as InfGraND and thus have identical inference times. These consistent results demonstrate InfGraND as an efficient and accurate method for graph learning.

### 5.2.4 Q4 - Ablation Study

We conduct an ablation study to evaluate the contribution of each core component in InfGraND. We use GCN and GraphSAGE as teacher models in the *transductive* and *inductive* settings, respectively, and evaluate performance on six benchmark datasets. To isolate the effect of each module, we consider two simplified variants: `w/Influence` and `w/Propagation`. In the `w/Influence` variant, the student MLP is trained using raw input features $\mathbf{X}$, without the multi-hop propagated version $\tilde{\mathbf{X}}$. In contrast, the `w/Propagation` variant disables the influence-guided objectives by setting the corresponding loss weights to zero: $\gamma_2 = 0$ and $\delta_2 = 0$ in $\mathcal{L}_s$ and $\mathcal{L}_d$, respectively. Results are summarized in Table 2.

Table 2: Ablation study comparing the full InfGraND model with variants using only influence guidance (w/Influence) or only feature propagation (w/Propagation). Results report classification accuracy (%) across multiple datasets under both inductive and transductive settings. Boldface and underline denote the best and second-best performance, respectively.

| Setting | Method | Cora | Citeseer | Pubmed | Amazon-Photo | Coauthor-CS | Coauthor-Phy |
|---------|--------|------|----------|--------|--------------|-------------|--------------|
| Transductive | Vanilla GCN | $82.2 \pm 0.6$ | $71.6 \pm 0.2$ | $79.2 \pm 0.3$ | $90.7 \pm 0.3$ | $89.3 \pm 0.0$ | $91.9 \pm 1.3$ |
| | Vanilla MLP | $57.8 \pm 1.0$ | $60.5 \pm 0.7$ | $72.8 \pm 0.4$ | $79.0 \pm 1.0$ | $87.8 \pm 0.5$ | $89.5 \pm 2.0$ |
| | GLNN | $83.1 \pm 0.3$ | $73.0 \pm 0.5$ | $79.4 \pm 0.6$ | $92.3 \pm 0.5$ | $92.6 \pm 0.4$ | $93.6 \pm 1.1$ |
| | InfGraND w/Influence | $83.4 \pm 0.8$ | $74.6 \pm 0.6$ | $\underline{81.1 \pm 0.4}$ | $92.1 \pm 0.2$ | $\underline{93.2 \pm 0.7}$ | $93.8 \pm 0.8$ |
| | InfGraND w/Propagation | $\underline{83.6 \pm 0.5}$ | $\underline{75.0 \pm 0.9}$ | $81.0 \pm 0.4$ | $\underline{93.0 \pm 0.3}$ | $93.0 \pm 0.7$ | $\underline{94.3 \pm 0.3}$ |
| | InfGraND (Full Model) | $\mathbf{84.0 \pm 0.5}$ | $\mathbf{75.2 \pm 1.1}$ | $\mathbf{81.3 \pm 0.2}$ | $\mathbf{94.2 \pm 0.4}$ | $\mathbf{93.5 \pm 0.6}$ | $\mathbf{94.7 \pm 0.0}$ |
| Inductive | Vanilla SAGE | $79.6 \pm 1.5$ | $64.7 \pm 0.8$ | $73.0 \pm 2.0$ | $\underline{91.2 \pm 0.8}$ | $89.0 \pm 0.7$ | $90.5 \pm 1.7$ |
| | Vanilla MLP | $58.4 \pm 2.5$ | $55.1 \pm 1.0$ | $70.8 \pm 1.2$ | $78.6 \pm 1.2$ | $88.1 \pm 1.3$ | $89.0 \pm 0.9$ |
| | GLNN | $69.6 \pm 1.7$ | $64.0 \pm 1.1$ | $72.4 \pm 0.5$ | $85.0 \pm 2.2$ | $89.3 \pm 0.7$ | $91.0 \pm 3.0$ |
| | InfGraND w/Influence | $70.5 \pm 1.6$ | $63.2 \pm 1.0$ | $74.0 \pm 1.1$ | $85.0 \pm 2.2$ | $90.8 \pm 1.5$ | $90.6 \pm 3.0$ |
| | InfGraND w/Propagation | $\underline{79.5 \pm 1.6}$ | $\underline{67.4 \pm 1.8}$ | $\underline{74.1 \pm 1.5}$ | $89.3 \pm 2.2$ | $\underline{91.4 \pm 1.1}$ | $\underline{92.6 \pm 2.7}$ |
| | InfGraND (Full Model) | $\mathbf{79.9 \pm 0.6}$ | $\mathbf{67.7 \pm 1.1}$ | $\mathbf{74.3 \pm 1.1}$ | $\mathbf{93.5 \pm 1.6}$ | $\mathbf{91.8 \pm 0.7}$ | $\mathbf{93.3 \pm 3.0}$ |

**Discussion.** Both the influence-guided objective and the feature propagation module independently contribute to the overall performance of InfGraND, and their effects are complementary. The `w/Influence` variant, which selectively transfers knowledge based on node influence within a graph, consistently improves upon the Vanilla MLP, GLNN, and teacher models across both transductive and inductive settings. Meanwhile, the `w/Propagation` variant equips the student with structural knowledge via pre-computed multi-hop features and yields particularly strong gains in the inductive setting, most notably on Citeseer, Cora, and Amazon-Photo. Specifically, it outperforms the Vanilla MLP by +10.7%, +12.3%, and +21.1% on these datasets, respectively. Importantly, these improvements come without introducing additional parameters or training overhead.

The significant gains observed on Cora, Citeseer, and Amazon-Photo in inductive setting can be attributed to their splitting characteristics (see Appendix A, Table 5). Citeseer, Cora, and Amazon-Photo exhibit higher proportions of observed and test nodes relative to the total node count. This broader coverage enables the propagation of features to capture more useful neighborhood information, thus improving the generalizability of the student model.

Note that both components are most effective when used together. In our experiments, all hyperparameters were tuned jointly for the full model. We observed that tuning them independently for the `w/Influence` or `w/Propagation` variants often yields better results than those reported in Table 2, as each variant has its own optimal configuration.

### 5.2.5 Q5 - Label-Scarce Setting

A key challenge in semi-supervised node classification is the high cost of labeling, which is inherently tedious, time-consuming, and resource-intensive. Often, we only have access to a limited number of labeled nodes. For example, in our main experiments, for the transductive setting, we use only 20 labeled nodes per class, which is comparatively very low compared to the number of test nodes (1000 nodes). This scarcity of labels, where $|\mathcal{V}_{\text{lab}}| << |\mathcal{V}_{\text{unl}}|$, highlights the need for models that perform comparatively well even with limited

labeled data. To evaluate InfGraND's performance in this setting, we conduct an experiment by limiting the number of labeled nodes used in training phase. We compare InfGraND and GLNN using a GraphSAGE teacher in transductive settings and three datasets: Cora, Citeseer, and Pubmed. In this experiment, we randomly selected 2, 4, and 8 labeled nodes per class, corresponding to 10%, 20%, and 40% of the original training set, respectively. We keep the testing set the same across all training settings. We use the same seed to ensure a fair comparison between the methods.

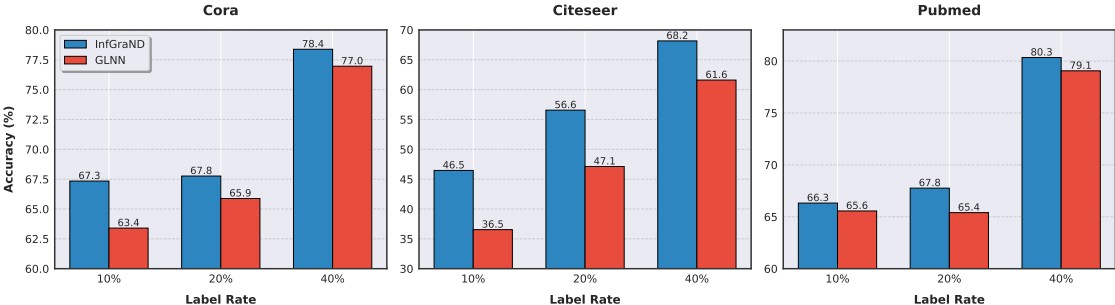

Figure 5: Accuracy comparison between InfGraND and GLNN under different proportions of labeled training samples (10%, 20%, 40%) on Cora, Citeseer, and PubMed, using GraphSAGE as the teacher model.

**Discussion.** The results, as shown in Figure 5, demonstrate that InfGraND consistently outperforms GLNN in all three test cases and datasets. InfGraND surpasses GLNN by an average of 4.17%. This superior performance under extreme label scarcity suggests that InfGraND's influence-guided objective effectively prioritizes influential nodes during training, enabling robust generalization even when labeled data are minimal.

### 5.2.6 Q6 - Hyperparameter Analysis

To better understand the behavior of InfGraND, we conduct a hyperparameter sensitivity analysis using a GraphSAGE teacher on Cora and Citeseer. We vary $\gamma_2$, $\delta_2$, and $\lambda$ across 10 values in $[0.0, 1.0]$, adjust the number of propagation steps $P$ from 1 to 4, and compare mean, max, and min pooling mechanisms. To emphasize relative trends rather than absolute performance, results on the Cora dataset are plotted after subtracting a constant offset of 10 percentage points.

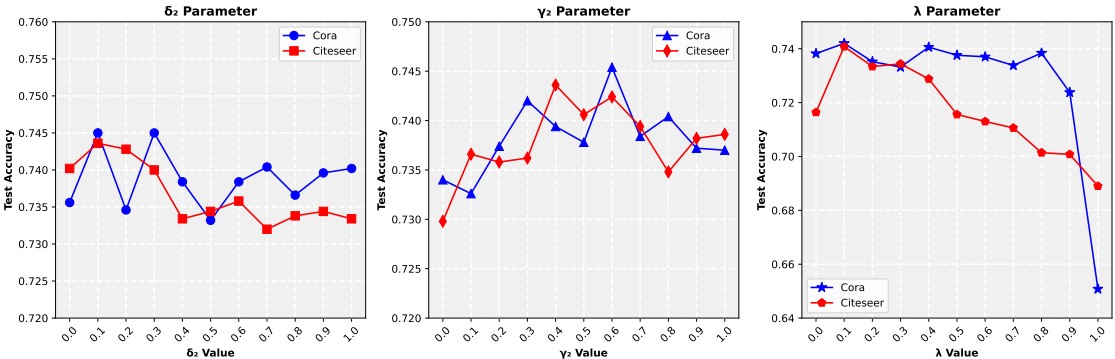

Figure 6: Sensitivity analysis of key parameters ($\delta_2$ (left), $\gamma_2$ (middle), and $\lambda$ (right)) on Cora and Citeseer datasets, showing their impact on test accuracy. $\delta_2$ controls influence-guided supervised loss, $\gamma_2$ governs influence-guided distillation loss, and $\lambda$ balances supervised and distillation losses.

**Effects of $\gamma_2$ and $\delta_2$.** The left and middle plots in Figure 6 illustrate the impact of the influence-guided terms in the supervised and distillation losses, controlled by $\delta_2$ and $\gamma_2$, respectively. The parameter $\delta_2$ appears in the influence-guided supervised loss $\mathcal{L}_s$ (Eq. 7), while $\gamma_2$ governs the influence-aware distillation loss $\mathcal{L}_d$ (Eq. 8). As shown in the left plot, setting $\delta_2 = 0.1$ yields better results than $\delta_2 = 0$ across datasets, with Cora showing an improvement of approximately 1%. However, increasing $\delta_2$ beyond this point does not consistently improve performance. Similarly, the middle plot shows that any non-zero value

of $\gamma_2$ improves performance over $\gamma_2 = 0$, suggesting that incorporating influence information, even with suboptimal parameters, is preferable to excluding it entirely.

**Sensitivity of $\lambda$.** The hyperparameter $\lambda \in [0,1]$ controls the trade-off between the supervised loss ($\mathcal{L}_s$) and the distillation loss ($\mathcal{L}_d$) in the total loss function (Eq. 9). A value of $\lambda = 1$ corresponds to pure supervised learning, while $\lambda = 0$ results in pure distillation. Importantly, $\lambda$ does not directly represent the proportion of knowledge transferred from the teacher versus the ground-truth labels; rather, it modulates the relative strength of their gradient contributions during optimization. The right plot in Figure 6 shows model performance as $\lambda$ varies. Performance peaks at $\lambda = 0.1$, indicating that the model benefits most when it receives a stronger gradient signal from the teacher soft labels than from the ground-truth hard labels. However, removing the supervised loss entirely ($\lambda = 0$) degrades performance, showing that while distillation provides the dominant learning signal, a degree of supervision remains beneficial. The worst results occur at $\lambda = 1.0$, where the model relies exclusively on labeled data. This trend is consistent with Table 1, where the MLP trained solely with supervised loss performs significantly worse than its distilled counterpart.

Table 3: Ablation study on features aggregation strategies. Results show classification accuracy (%) with different numbers of neighborhood hops.

| Dataset | $P$ Hops | Mean | Maximum | Minimum |
|---|---|---|---|---|
| Cora | 1-hop | $82.24 \pm 0.6$ | $83.82 \pm 0.5$ | $\mathbf{82.30 \pm 1.0}$ |
| | 2-hop | $\mathbf{84.50 \pm 0.6}$ | $\mathbf{84.18 \pm 0.2}$ | $82.70 \pm 0.7$ |
| | 3-hop | $84.26 \pm 0.5$ | $83.90 \pm 0.5$ | $81.58 \pm 1.0$ |
| | 4-hop | $83.62 \pm 0.7$ | $83.94 \pm 0.7$ | $81.22 \pm 0.9$ |
| Citeseer | 1-hop | $73.16 \pm 1.0$ | $73.32 \pm 0.4$ | $\mathbf{71.22 \pm 2.3}$ |
| | 2-hop | $\mathbf{74.02 \pm 1.0}$ | $\mathbf{73.50 \pm 0.9}$ | $70.10 \pm 4.1$ |
| | 3-hop | $73.38 \pm 1.3$ | $73.62 \pm 0.8$ | $69.62 \pm 3.7$ |
| | 4-hop | $73.08 \pm 0.7$ | $72.90 \pm 0.6$ | $68.92 \pm 6.0$ |
| Pubmed | 1-hop | $80.56 \pm 0.3$ | $80.24 \pm 0.1$ | $80.74 \pm 0.3$ |
| | 2-hop | $\mathbf{81.16 \pm 0.4}$ | $80.28 \pm 0.2$ | $80.58 \pm 0.4$ |
| | 3-hop | $80.80 \pm 0.9$ | $\mathbf{80.44 \pm 0.5}$ | $80.46 \pm 0.5$ |
| | 4-hop | $80.96 \pm 0.2$ | $80.30 \pm 0.7$ | $\mathbf{80.88 \pm 0.7}$ |

**Pooling and P.** For information propagation, as defined in Eq. 6, there are two design choices: the number of propagation steps $P$, and the pooling mechanism. Table 3 shows that averaging features from 2-hop neighborhoods yields the best performance across Cora ($84.50 \pm 0.6\%$), Citeseer ($74.02 \pm 1.0\%$), and Pubmed ($81.16 \pm 0.4\%$). The '*minimum*' aggregation consistently performs worst, likely due to information loss during feature propagation. Extending the neighborhood beyond 2-hops does not lead to further improvements, suggesting that distant neighbors may introduce noise rather than useful structure.

# 6 Conclusion and Future Work

This work advances GNN-to-MLP distillation by challenging the uniform treatment of nodes in the distillation process. We define and compute node influence scores and show that prioritizing high-influence nodes improves the generalization of GNNs. Building on this insight, we introduce InfGraND, which distills influence-guided knowledge from a teacher GNN to an MLP student. The student also leverages a one-time feature propagation step, inspired by industrial practices such as storing embeddings in lookup tables. Experiments across seven homophily datasets confirm InfGraND's superiority. Across six datasets and three teacher architectures, InfGraND improves over vanilla MLPs by 12.6% (transductive) and 9.3% (inductive), while also surpassing its GNN teachers by 3.2% and 2.6%, respectively. It also demonstrates clear advantages over prior distillation methods, including FF-G2M, KRD, and HGMD. Additionally, on the large-scale OGBN-Arxiv dataset, InfGraND improves over MLPs by 19.5% (transductive) and 9.5% (inductive), and outperforms KRD by 0.4% on average over the two settings. We also conduct a diverse set of experiments to provide insights into the model's behavior from different angles and in various scenarios. These results highlight InfGraND's strong performance and its potential for practical deployment of models that leverage graph structure. Future work will involve extending our method to broader applications, including different graph types such as heterophilous and dynamic graphs. We also plan to investigate hybrid methods that combine entropy-based discrimination with structural-aware approaches.

**Acknowledgments**

This work was undertaken thanks in part to funding from the Connected Minds program, supported by the Canada First Research Excellence Fund, grant #CFREF-2022-00010, and the New Frontiers in Research Fund, grant #NFRFE-2022-00197.

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

# Appendix

## A Dataset Statistics

In this study, we evaluate our approach on seven widely used datasets for graph analysis: three small-scale citation datasets, Cora (Sen et al., 2008), Citeseer (Giles et al., 1998), and Pubmed (McCallum et al., 2000); three large-scale datasets, Coauthor-CS, Coauthor-Phy, and Amazon-Photo (Shchur et al., 2018); and the OGBN-arxiv dataset (Hu et al., 2020); a large-scale benchmark from the Open Graph Benchmark (OGB) collection. Table 4 summarizes the detailed statistics for all datasets used in our experiments.

Table 4: General dataset statistics.

| Dataset | #Nodes | #Edges | #Features | #Classes | Label Rate |
|---|---|---|---|---|---|
| Cora | 2,708 | 5,278 | 1,433 | 7 | 5.2% |
| Citeseer | 3,327 | 4,614 | 3,703 | 6 | 3.6% |
| Pubmed | 19,717 | 44,324 | 500 | 3 | 0.3% |
| Amazon-Photo | 7,650 | 119,081 | 745 | 8 | 2.1% |
| Coauthor-CS | 18,333 | 81,894 | 6,805 | 15 | 1.6% |
| Coauthor-Phy | 34,493 | 247,962 | 8,415 | 5 | 0.3% |
| OGBN-arxiv | 169,343 | 1,166,243 | 128 | 40 | 53.7% |

Table 5: Splitting statistics for inductive evaluation.

| Dataset | #Total | #Obs. | #Test | Obs. (%) | Test (%) | Obs/Test |
|---|---|---|---|---|---|---|
| Cora | 2708 | 1440 | 200 | 53.18% | 7.39% | 7.20 |
| Citeseer | 3327 | 1420 | 200 | 42.68% | 6.01% | 7.10 |
| Pubmed | 19717 | 1360 | 200 | 6.90% | 1.01% | 6.80 |
| Coauthor-CS | 18333 | 1600 | 200 | 8.73% | 1.09% | 8.00 |
| Coauthor-Phy | 34493 | 1400 | 200 | 4.06% | 0.58% | 7.00 |
| Amazon-Photo | 7650 | 1460 | 200 | 19.08% | 2.61% | 7.30 |
| OGBN-arxiv | 169343 | 164483 | 4860 | 97.13% | 2.87% | 33.85 |

Table 5 reports node-level splitting statistics used in the inductive setting. We observe a notable variation in the ratio of observed and test nodes to the total number of nodes. For example, Cora and Citeseer have a relatively high percentage of observed nodes (over 40%), which facilitates effective feature propagation during distillation and inference. In contrast, Pubmed and Coauthor-Phy exhibit sparse supervision (under 7% observed), making generalization more challenging. These variations in data availability directly affect the model's ability to learn transferable representations in the inductive setting.

## B Hyperparameter Settings and Tuning

The model architectures were built using 2-4 layers, with the hidden dimension searched over the set $\{128, 256, 512, 1024, 2048\}$. For optimization, the learning rate was tuned from $\{0.001, 0.005, 0.01\}$ and the weight decay was selected from $\{0.0, 5 \times 10^{-4}\}$. All models were trained for a maximum of 500 epochs, utilizing an early stopping criterion that halts training if validation accuracy does not improve for 50 consecutive epochs. The distillation temperature $\tau$ was selected from the range $[0.5, 2.0]$, and the knowledge distillation weight $\lambda$ was chosen from the discrete set $\{0.0, 0.1, 0.2, 0.3, 0.5\}$. Furthermore, dropout rates for both teacher and student models were adjusted in the set $\{0.0, 0.1, \ldots, 0.8\}$. For the influence-guided weights $(\delta_1, \gamma_1, \delta_2, \gamma_2)$, the parameters $\delta_1$ and $\gamma_1$ were selected from the set $\{0.001, 0.01, 0.1, 0.4, 0.5, 0.6, 0.8, 0.9, 1.0\}$, while

Table 6: Hyperparameters for InfGraND in transductive and inductive settings, including the distillation weight ($\lambda$) and influence-guided coefficients ($\delta_1$, $\delta_2$, $\gamma_1$, $\gamma_2$).

| Dataset | Teacher | Transductive | | | | | Inductive | | | | |
|---|---|---|---|---|---|---|---|---|---|---|---|
| | | $\lambda$ | $\delta_1$ | $\delta_2$ | $\gamma_1$ | $\gamma_2$ | $\lambda$ | $\delta_1$ | $\delta_2$ | $\gamma_1$ | $\gamma_2$ |
| Cora | GCN | 0.1 | 0.6 | 0.2 | 0.8 | 0.4 | 0.1 | 1.0 | 0.2 | 0.8 | 0.3 |
| | SAGE | 0.5 | 0.8 | 0.1 | 1.0 | 0.4 | 0.5 | 1.0 | 0.2 | 1.0 | 0.2 |
| | GAT | 0.5 | 0.4 | 0.2 | 1.0 | 0.1 | 0.5 | 0.9 | 0.8 | 1.0 | 0.2 |
| Citeseer | GCN | 0.0 | 0.6 | 0.1 | 0.6 | 0.4 | 0.0 | 1.0 | 0.2 | 0.8 | 0.3 |
| | GAT | 0.1 | 0.8 | 0.2 | 0.6 | 0.2 | 0.1 | 0.9 | 0.8 | 1.0 | 0.2 |
| | SAGE | 0.1 | 0.6 | 0.1 | 0.4 | 0.4 | 0.0 | 1.0 | 0.2 | 1.0 | 0.2 |
| Pubmed | GCN | 0.0 | 1.0 | 0.1 | 0.6 | 0.4 | 0.5 | 1.0 | 0.2 | 0.2 | 1.0 |
| | GAT | 0.0 | 0.4 | 0.2 | 0.4 | 0.1 | 0.0 | 0.5 | 0.2 | 0.2 | 1.0 |
| | SAGE | 0.0 | 0.8 | 0.1 | 0.8 | 0.2 | 0.0 | 0.1 | 0.1 | 0.8 | 0.1 |
| Amazon | GCN | 0.2 | 1.0 | 0.2 | 0.8 | 0.1 | 0.2 | 1.0 | 0.1 | 1.0 | 0.1 |
| | GAT | 0.3 | 0.6 | 0.1 | 0.8 | 0.01 | 0.3 | 0.8 | 0.1 | 0.6 | 0.1 |
| | SAGE | 0.2 | 0.4 | 0.7 | 0.7 | 0.4 | 0.2 | 0.9 | 0.4 | 0.6 | 0.1 |
| CS | GCN | 0.3 | 1.0 | 0.1 | 0.5 | 0.5 | 0.3 | 1.0 | 0.2 | 1.0 | 0.2 |
| | GAT | 0.0 | 1.0 | 0.1 | 0.01 | 0.01 | 0.3 | 0.8 | 0.1 | 0.6 | 0.1 |
| | SAGE | 0.5 | 0.5 | 0.5 | 1.0 | 0.1 | 0.5 | 0.4 | 0.4 | 0.4 | 0.1 |
| Physics | GCN | 0.2 | 0.1 | 0.01 | 0.1 | 0.1 | 0.2 | 0.1 | 0.001 | 0.6 | 0.001 |
| | GAT | 0.2 | 0.8 | 0.8 | 0.8 | 0.8 | 0.2 | 1.0 | 0.6 | 0.1 | 0.001 |
| | SAGE | 0.5 | 1.0 | 0.2 | 0.2 | 0.2 | 0.5 | 0.1 | 0.01 | 0.4 | 0.01 |

$\delta_2$ and $\gamma_2$ were selected from $\{0.0001, 0.001, 0.01, 0.05, 0.1, 0.2, 0.4, 0.5, 0.6, 0.8, 1.0\}$. For OGBN-Arxiv, we used edge dropout during distillation, tuning the drop rate from $\{0.85, 0.90, 0.95\}$. The final selected values for the five core InfGraND hyperparameters ($\lambda$, $\delta_1$, $\delta_2$, $\gamma_1$, $\gamma_2$) are reported in Table 6.

## C Visualization of Class Separation

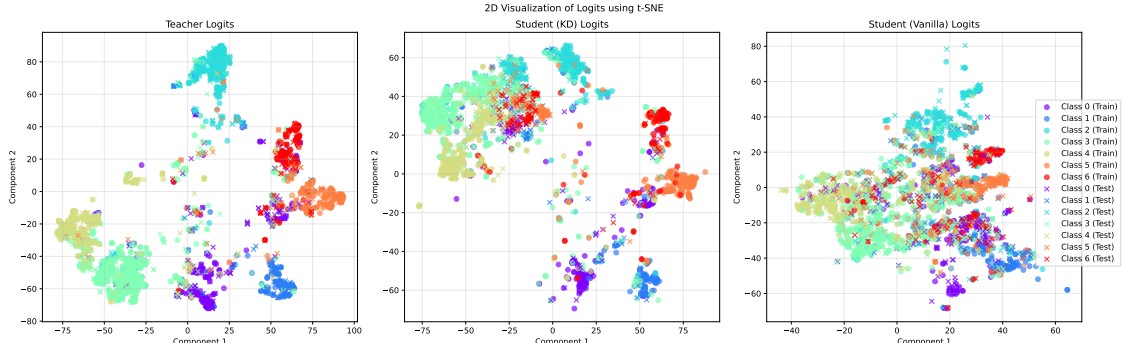

Figure 7: t-SNE visualization of 7-dimensional logits reduced to 2D from the Cora dataset under the inductive setting. The plots compare the representation spaces of the Teacher (left), Student KD (middle), and Vanilla Student (right) models. Circles denote training samples and crosses denote test samples, with colors indicating class membership.

Table 7 and Figure 7 illustrate the effect of knowledge distillation on the logit vector of the student model. The teacher model (left plot in Figure 7) shows clear and distinct clusters with a high silhouette score (0.243) and large mean intercluster distance (80.3), which is also apparent in the t-SNE plot where classes are well separated. The knowledge-distilled student (the middle plot in Figure 7), al-

Table 7: Model comparison metrics across different evaluation criteria. Bold values indicate best performance in each metric. CH: Calinski-Harabasz, DB: Davies-Bouldin.

| Model | CE | | Cluster Quality | | |
|---|---|---|---|---|---|
| | Train | Test | Silhouette | CH Score | DB Score |
| Teacher | **0.015** | 0.636 | **0.243** | **1215.0** | **1.78** |
| InfGraND | 0.160 | **0.585** | 0.002 | 360.1 | 4.39 |
| MLP | 0.157 | 1.389 | -0.021 | 262.7 | 6.49 |

though exhibiting less pronounced clustering (silhouette score of 0.002 and mean distance of 50.9), maintains a structural pattern similar to the teacher and achieves a lower test cross-entropy (0.585 versus 0.636), indicating that it learns a more efficient and generalized representation rather than merely replicating the teacher's exact cluster boundaries. In contrast, the vanilla student (right plot in Figure 7) presents very poor clustering performance, as shown by its negative silhouette score (–0.021) and low mean distance (28.4), resulting in significant overlap among classes and a much higher test cross-entropy (1.389). Overall, these results suggest that knowledge distillation effectively transfers the teacher's structural knowledge to the student while promoting a representation space that is more conducive to generalization.

## D    Theoretical Analysis

We define the distillation loss $\mathcal{L}_d$ over a set of directed edges $\mathcal{E}$ among nodes in a graph, where node $i$ has source representation $\mathbf{h}_i^s$ and node $j$ has target representation $\mathbf{h}_j^t$. Given a set of representations $\{\mathbf{h}_k\}$, we denote the softmax probability vector as $\sigma(\mathbf{h}_k)$. Without loss of generality, we set $\tau = 1$; the loss is then given by:

$$\mathcal{L}_d = \frac{1}{|\mathcal{E}|} \left( \gamma_1 \sum_{(i,j) \in \mathcal{E}} D_{\mathrm{KL}}(\sigma(\mathbf{h}_i^s) \parallel \sigma(\mathbf{h}_j^t)) + \gamma_2 \sum_{(i,j) \in \mathcal{E}} \mathcal{I}_g(v_j) \cdot D_{\mathrm{KL}}(\sigma(\mathbf{h}_i^s) \parallel \sigma(\mathbf{h}_j^t)) \right), \qquad (10)$$

where $\gamma_1$ and $\gamma_2$ are scalar weights, and $\mathcal{I}_g(v_j) \in [0, 1]$ is the Global Influence Score of the neighbor node $v_j$, as defined in Eq. 5. $\mathcal{L}_d$ can be rewritten using local neighborhoods as:

$$\mathcal{L}_d = \sum_{i \in \mathcal{V}} \sum_{j \in \mathcal{N}(v_i)} (\gamma_1 + \gamma_2 \cdot \mathcal{I}_g(v_j)) \cdot \frac{1}{|\mathcal{N}(v_i)|} \cdot D_{\mathrm{KL}}(\sigma(\mathbf{h}_i^s) \parallel \sigma(\mathbf{h}_j^t)). \qquad (11)$$

The KL divergence is defined as:

$$D_{\mathrm{KL}}(\sigma(\mathbf{h}_i^s) \| \sigma(\mathbf{h}_j^t)) = \sigma(\mathbf{h}_i^s)^\top (\log \sigma(\mathbf{h}_i^s) - \log \sigma(\mathbf{h}_j^t)). \tag{12}$$

We denote the output of the student model by $\sigma(\mathbf{h}_i^s)$. The student model is a two-layer neural network with a ReLU activation function in the first layer. We assume that the output of the first layer is positive element-wise (i.e., $\mathbf{W}_1\mathbf{x}_i + \mathbf{b}_1 > 0$), since otherwise $\sigma(\mathbf{h}_i^s)$ would reduce to the bias of the second layer.

$$\sigma(\mathbf{h}_i^s) = \sigma(\mathbf{W}_2(\mathbf{W}_1\mathbf{x}_i + \mathbf{b}_1) + \mathbf{b}_2), \tag{13}$$

where:

$$\mathbf{x}_i \in \mathbb{R}^d, \quad \mathbf{W}_1 \in \mathbb{R}^{f \times d}, \quad \mathbf{b}_1 \in \mathbb{R}^f,$$
$$\mathbf{W}_2 \in \mathbb{R}^{c \times f}, \quad \mathbf{b}_2 \in \mathbb{R}^c.$$

### D.1 Gradient Derivation of $\mathcal{L}_d$

The distillation process involves training the student model to match the teacher's representation using the loss function defined in Eq. 10. The optimizer performs backpropagation based on this objective. Since the gradient signal dictates how the student model updates its parameters, analyzing this signal is essential to understanding the behavior of the proposed method.

Recalling that $\mathbf{h}_i^s$ is the student representation and $\sigma(\mathbf{h}_i^s)$ denotes its softmax output, the Jacobian of the softmax with respect to $\mathbf{h}_i^s$ is given by:

$$\frac{\partial \sigma(\mathbf{h}_i^s)}{\partial \mathbf{h}_i^s} = \mathrm{diag}(\sigma(\mathbf{h}_i^s)) - \sigma(\mathbf{h}_i^s)\,\sigma(\mathbf{h}_i^s)^\top \in \mathbb{R}^{c \times c}. \tag{14}$$

Using the chain rule, the gradients of the softmax output $\sigma(\mathbf{h}_i^s)$ with respect to the model parameters are:

$$\frac{\partial \sigma(\mathbf{h}_i^s)}{\partial \mathbf{b}_2} = \left(\mathrm{diag}(\sigma(\mathbf{h}_i^s)) - \sigma(\mathbf{h}_i^s)\sigma(\mathbf{h}_i^s)^\top\right) \in \mathbb{R}^{c \times c}, \tag{15}$$

$$\frac{\partial \sigma(\mathbf{h}_i^s)}{\partial \mathbf{b}_1} = \left(\mathrm{diag}(\sigma(\mathbf{h}_i^s)) - \sigma(\mathbf{h}_i^s)\sigma(\mathbf{h}_i^s)^\top\right) \mathbf{W}_2 \in \mathbb{R}^{c \times f}, \tag{16}$$

$$\frac{\partial \sigma(\mathbf{h}_i^s)}{\partial \mathbf{W}_2} = \left(\mathrm{diag}(\sigma(\mathbf{h}_i^s)) - \sigma(\mathbf{h}_i^s)\sigma(\mathbf{h}_i^s)^\top\right) (\mathbf{W}_1\mathbf{x}_i + \mathbf{b}_1)^\top \in \mathbb{R}^{c \times f}, \tag{17}$$

$$\frac{\partial \sigma(\mathbf{h}_i^s)}{\partial \mathbf{W}_1} = \left[\left(\mathrm{diag}(\sigma(\mathbf{h}_i^s)) - \sigma(\mathbf{h}_i^s)\sigma(\mathbf{h}_i^s)^\top\right) \mathbf{W}_2\right] \otimes \mathbf{x}_i^\top \in \mathbb{R}^{c \times f \times d}. \tag{18}$$

While Eqs. 15–18 describe how the softmax output depends on the model parameters, the actual learning signal during distillation originates from the loss function. To analyze how this signal propagates, we need to derive the gradient of $\mathcal{L}_d$ with respect to the model parameters. We first differentiate $\mathcal{L}_d$ with respect to the softmax output, then apply the chain rule to obtain parameter gradients.

$$\nabla \mathcal{L}_d = \sum_{i \in \mathcal{V}} \sum_{j \in \mathcal{N}(v_i)} (\gamma_1 + \gamma_2 \cdot \mathcal{I}_g(v_j)) \cdot \frac{1}{|\mathcal{N}(v_i)|} \cdot D'_{\mathrm{KL}}(\sigma(\mathbf{h}_i^s) \| \sigma(\mathbf{h}_j^t)), \tag{19}$$

which can be further written as:

$$\nabla \mathcal{L}_d = \sum_{i \in \mathcal{V}} \sum_{j \in \mathcal{N}(v_i)} \frac{(\gamma_1 + \gamma_2 \cdot \mathcal{I}_g(v_j))}{|\mathcal{N}(v_i)|} \cdot (\nabla \sigma(\mathbf{h}_i^s))^\top \cdot \left[\log \sigma(\mathbf{h}_i^s) - \log \sigma(\mathbf{h}_j^t) + \mathbf{1}\right]. \tag{20}$$

Incorporating Eqs. 15–18 into the formulation of Eq. 20, we obtain:

$$
\nabla_{\mathbf{b}_2}\mathcal{L}_d \in \mathbb{R}^c = \sum_{i\in\mathcal{V}}\sum_{j\in\mathcal{N}(v_i)}\frac{\gamma_1 + \gamma_2 \cdot \mathcal{I}_g(v_j)}{|\mathcal{N}(v_i)|} \tag{21}
$$
$$
\cdot \Big( \mathrm{diag}(\sigma(\mathbf{h}_i^s)) - \sigma(\mathbf{h}_i^s)\sigma(\mathbf{h}_i^s)^\top \Big)
$$
$$
\cdot \Big[ \log\sigma(\mathbf{h}_i^s) - \log\sigma(\mathbf{h}_j^t) + \mathbf{1} \Big],
$$

$$
\nabla_{\mathbf{b}_1}\mathcal{L}_d \in \mathbb{R}^f = \sum_{i\in\mathcal{V}}\sum_{j\in\mathcal{N}(v_i)}\frac{(\gamma_1 + \gamma_2 \cdot \mathcal{I}_g(v_j))}{|\mathcal{N}(v_i)|} \tag{22}
$$
$$
\cdot \Big( (\mathrm{diag}(\sigma(\mathbf{h}_i^s)) - \sigma(\mathbf{h}_i^s)\sigma(\mathbf{h}_i^s)^\top)\mathbf{W}_2 \Big)
$$
$$
\cdot \Big[ \log\sigma(\mathbf{h}_i^s) - \log\sigma(\mathbf{h}_j^t) + 1 \Big],
$$

$$
\nabla_{\mathbf{W}_2}\mathcal{L}_d \in \mathbb{R}^{c\times f} = \sum_{i\in\mathcal{V}}\sum_{j\in\mathcal{N}(v_i)}\frac{\gamma_1 + \gamma_2 \cdot \mathcal{I}_g(v_j)}{|\mathcal{N}(v_i)|} \tag{23}
$$
$$
\cdot \Big( \big( \mathrm{diag}(\sigma(\mathbf{h}_i^s)) - \sigma(\mathbf{h}_i^s)\sigma(\mathbf{h}_i^s)^\top \big)(\mathbf{W}_1\mathbf{x}_i + \mathbf{b}_1)^\top \Big)
$$
$$
\cdot \Big[ \log\sigma(\mathbf{h}_i^s) - \log\sigma(\mathbf{h}_j^t) + \mathbf{1} \Big],
$$

$$
\nabla_{\mathbf{W}_1}\mathcal{L}_d \in \mathbb{R}^{f\times d} = \sum_{i\in\mathcal{V}}\sum_{j\in\mathcal{N}(v_i)}\frac{\gamma_1 + \gamma_2 \cdot \mathcal{I}_g(v_j)}{|\mathcal{N}(v_i)|} \tag{24}
$$
$$
\cdot \Big( \big[ \big( \mathrm{diag}(\sigma(\mathbf{h}_i^s)) - \sigma(\mathbf{h}_i^s)\sigma(\mathbf{h}_i^s)^\top \big)\mathbf{W}_2 \big] \otimes \mathbf{x}_i^\top \Big)
$$
$$
\cdot \Big[ \log\sigma(\mathbf{h}_i^s) - \log\sigma(\mathbf{h}_j^t) + \mathbf{1} \Big],
$$

These expressions describe how the distillation loss $\mathcal{L}_d$ backpropagates gradients to the model parameters.

### D.2  Effect of $I(j)$

In Eq. 20, $\gamma_1$ and $\gamma_2$ are scalar coefficients in $(0,1]$, and $\mathcal{I}_g(v_j)$ is the GIS of the neighbor node $v_j$. This gradient expression reveals two key multiplicative components:

- The term $(\gamma_1 + \gamma_2 \cdot \mathcal{I}_g(v_j))$ serves as a scalar weight that modulates the contribution of each neighbor to the gradient. A higher $\mathcal{I}_g(v_j)$ increases the influence of that neighbor on the gradient update.

- A directional term $\big[ \log\sigma(\mathbf{h}_i^s) - \log\sigma(\mathbf{h}_j^t) + \mathbf{1} \big]$, which captures the distributional divergence between the predictions of the student and the teacher.

This decomposition shows that $\mathcal{I}_g(v_j)$ acts as an *amplifier*, strengthening the gradient signal and helping the student better identify and correct its mistakes. For example, consider node $i = 1$ and $j = 2$. Assuming $\gamma_1 = \gamma_2 = 1$, the gradient simplifies to:

$$
\nabla\mathcal{L}_d^1 = \frac{(1 + \mathcal{I}_g(v_2))}{|\mathcal{N}(1)|} \cdot (\nabla\sigma(\mathbf{h}_1^s))^\top \cdot \big[ \log\sigma(\mathbf{h}_1^s) - \log\sigma(\mathbf{h}_2^t) + \mathbf{1} \big]. \tag{25}
$$

The magnitude of the equation in 25 is given by:

$$\left\|\nabla\mathcal{L}_d^1\right\| = \frac{1+\mathcal{I}_g(v_2)}{|\mathcal{N}(1)|}\cdot\left\|(\nabla\sigma(\mathbf{h}_1^s))^\top\cdot\left[\log\sigma(\mathbf{h}_1^s)-\log\sigma(\mathbf{h}_2^t)+\mathbf{1}\right]\right\|. \tag{26}$$

In Eq. 26, $\mathcal{I}_g(v_2)\in[0,1]$ scales the gradient magnitude of the distillation loss, assigning greater weight to more influential neighbors in the update. This mechanism encourages the student model to align more closely with informative neighbors during distillation.

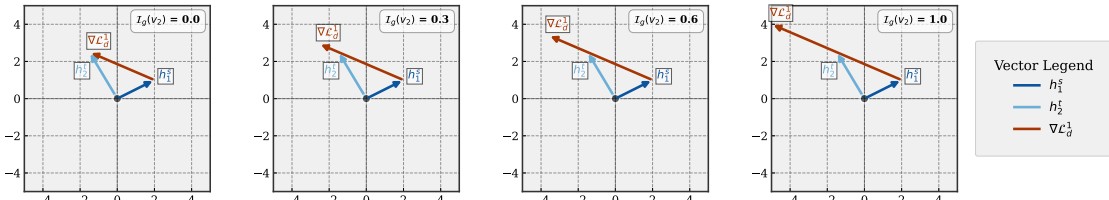

Figure 8: **Effect of influence score $\mathcal{I}_g(v_j)$ on gradient magnitude in KD.** The diagrams show how increasing $I(v_2)\in\{0.0,0.3,0.6,1.0\}$ scales the magnitude of the gradient vector. Higher scores amplify the correction signal for semantically important neighbors.

As an example to illustrate Equation 26, consider a scenario with two classes, where the dimension of the logits is 2. In this case, the student and teacher outputs can be represented as 2D vectors. As shown in Figure 8, the blue and light blue vectors correspond to the logits of nodes 1 and 2, respectively. The red vector labeled "$\nabla\mathcal{L}_d^1$" is approximately the gradient signal, which guides the representation of node 1 to align with that of node 2. Starting from the leftmost subfigure, we observe how the influence score $\mathcal{I}_g(v_2)\in[0,1]$ scales the magnitude of the gradient of the distillation loss. As $\mathcal{I}_g(v_2)$ increases across the subfigures (e.g., 0.3, 0.6, 1.0), the magnitude of the gradient vector increases proportionally, amplifying the gradient update. This effect encourages the model to focus on aligning with more informative neighbors during distillation. The same scaling behavior generalizes to Equations 21, 22, 23, and 24.

### D.3 The Necessity of $\gamma_1$

The design of the distillation loss encourages the student model to learn more from high-influence nodes. However, low-influence nodes can also provide valuable knowledge to the student. If $\gamma_1$ were omitted, the gradients from low-influence nodes would be suppressed during distillation, as $\mathcal{I}_g(v_j)$ would shrink the gradient signal entirely for those nodes.

For example, in the gradient expression of Eq. 20, when $\mathcal{I}_g(v_j)=0$, the update reduces to:

$$\nabla\mathcal{L}_d^{(\mathcal{I}_g(v_j)=0)} = \sum_{i\in\mathcal{V}}\sum_{j\in\mathcal{N}(v_i)}\frac{\gamma_1}{|\mathcal{N}(v_i)|}\cdot(\nabla\sigma(\mathbf{h}_i^s))^\top\cdot\left[\log\sigma(\mathbf{h}_i^s)-\log\sigma(\mathbf{h}_j^t)+\mathbf{1}\right], \tag{27}$$

which remains a meaningful gradient signal aligned with the teacher prediction $\sigma(\mathbf{h}_j^t)$. Thus, $\gamma_1$ plays a foundational role in preserving knowledge transfer from all neighbors, while $\gamma_2\cdot\mathcal{I}_g(v_j)$ provides additional fine-grained emphasis based on learned importance.

## E  Influence Score vs. Degree/PageRank

Here, we conducted a focused ablation study comparing our influence metric against normalized Degree and PageRank centrality measures. To isolate the impact of the weighting metric itself, we disabled the feature propagation module (Section 4.2) and used only the influence-guided loss component of our framework. We

then substituted our influence score $\mathcal{I}_g(v_i)$ with normalized Degree and PageRank scores within Eq. 7 and 8. This controlled comparison isolates the node importance metric as the sole variable. We independently tuned the core hyperparameters $(\gamma_1, \gamma_2, \delta_1, \delta_2)$, using validation accuracy as the optimization criterion. All experiments were conducted in the transductive setting across three teacher architectures (GCN, GAT, GraphSAGE) and three benchmark datasets (Cora, Citeseer, Pubmed).

Table 8 presents the comparative results across all teacher-dataset configurations. Our proposed influence metric consistently outperforms Degree and PageRank Bojchevski et al. (2019) in all nine experimental settings, achieving average improvements of +2.3% and +2.7%, respectively. This performance advantage arises from a fundamental difference in what is being measured. Degree and PageRank are purely link-analysis algorithms that only consider how nodes are connected, completely ignoring node features. In contrast, our influence metric considers both node features and their connections in the graph. This dual consideration of features and topology makes it fundamentally better suited for guiding knowledge distillation in GNNs, where both aspects are critical.

Table 8: Comparison of node importance metrics for knowledge distillation guidance (transductive setting). Bold values indicate the best performance.

| Teacher | Dataset | KD w/Influence | KD w/Degree | KD w/PageRank |
|---------|---------|----------------|-------------|---------------|
| **GCN** | Cora | **83.4 ± 0.8** | 81.1 ± 0.9 | 80.8 ± 0.7 |
| | Citeseer | **74.6 ± 0.6** | 71.6 ± 0.3 | 70.7 ± 0.8 |
| | Pubmed | **81.1 ± 0.4** | 79.7 ± 0.0 | 79.5 ± 0.2 |
| **GAT** | Cora | **84.0 ± 0.0** | 81.6 ± 0.7 | 79.7 ± 0.9 |
| | Citeseer | **72.2 ± 1.9** | 69.3 ± 5.2 | 69.0 ± 5.0 |
| | Pubmed | **80.6 ± 0.4** | 79.0 ± 0.6 | 79.8 ± 0.4 |
| **SAGE** | Cora | **84.4 ± 0.0** | 81.5 ± 1.3 | 81.1 ± 0.8 |
| | Citeseer | **72.7 ± 1.1** | 70.2 ± 2.0 | 69.5 ± 2.5 |
| | Pubmed | **80.3 ± 0.4** | 78.0 ± 1.7 | 79.9 ± 0.3 |

## F  Influence Computation: Complexity and Scalability Analysis

Our influence computation method has two implementations with distinct computational characteristics. The naive all-pairs approach computes cosine similarity between all node pairs, requiring $O(N^2 \cdot D)$ time complexity and $O(N^2)$ space complexity, where $N$ is the number of nodes and $D$ is the feature dimension. Although this provides the most complete influence profile, it becomes infeasible for large graphs, OGBN-Arxiv. The k-hop neighborhood approximation restricts influence computation to nodes within k hops, requiring $O(|E_k| \cdot D)$ time and $O(|E_k|)$ space, where $|E_k|$ denotes edges within k-hop neighborhoods. For sparse graphs with $k = 2$, this is approximately $O(N \cdot \bar{d}^2 \cdot D)$ where $\bar{d}$ is the average degree. Since $|E_k| \ll N^2$ for sparse graphs, this significantly reduces memory usage. The k-hop approximation is principled for large-scale graphs. In large-scale graphs, a node's influence decays with distance. For nodes many hops away, the influence score approaches zero. Therefore, the k-hop approximation captures meaningful local influence while discarding negligible distant interactions. Both implementations use batched GPU computation for memory efficiency.

We conducted timing experiments across seven benchmark datasets using GPU-batched implementation on a single NVIDIA GPU (32GB VRAM). Table 9 presents the results. For the six small to medium datasets (Cora, Citeseer, Pubmed, Amazon-Photo, Coauthor-CS, Coauthor-Phy), we use the naive all-pairs approach as it is computationally affordable and provides exact influence scores without approximation. For OGBN-Arxiv (169K nodes, 2.5M edges), we use the 2-hop approximation, which completes in 70 seconds (1.2 minutes) with 0.41 milliseconds per node. The naive approach for OGBN-Arxiv is infeasible due to memory constraints. These results demonstrate that our method scales efficiently from 2.7K to 169K nodes.

Table 9: Influence computation runtime (GPU-batched, single 32GB GPU). Times reported in seconds. *Naive approach for OGBN-Arxiv requires ~114 GB memory (infeasible).

| Dataset | Nodes | Edges | Naive (s) | 1-hop (s) | 2-hop (s) | 3-hop (s) |
|---------|-------|-------|-----------|-----------|-----------|-----------|
| Cora | 2,708 | 13,264 | 5.68 | 0.95 | 1.18 | 2.34 |
| Citeseer | 3,327 | 12,431 | 0.13 | 1.29 | 1.60 | 2.11 |
| Pubmed | 19,717 | 108,365 | 5.25 | 9.69 | 13.48 | 34.06 |
| Amazon-Photo | 7,650 | 245,812 | 0.89 | 3.75 | 22.93 | 202.60 |
| Coauthor-CS | 18,333 | 182,121 | 5.39 | 9.42 | 14.70 | 63.28 |
| Coauthor-Phy | 34,493 | 530,417 | 21.95 | 24.16 | 47.82 | 354.43 |
| OGBN-Arxiv | 169,343 | 2,501,829 | N/A* | 49.78 | 70.13 | 498.46 |

Influence computation is a one-time preprocessing step during training, not a recurring cost during inference. Once computed, influence scores are used to weight the distillation loss but do not affect the student

MLP's deployment. Therefore, even if computation takes minutes, it does not impact deployment latency or real-time prediction performance. Our influence computation method is computationally efficient (under 2 minutes for 169K nodes) and practical (one-time preprocessing with zero inference overhead).

## G  InfGraND Algorithm

Algorithm 1 presents the complete InfGraND training procedure, which consists of three main phases: (1) training the teacher GNN on the graph structure, (2) generating teacher predictions for all nodes, and (3) training the student MLP using the influence-guided distillation approach. The algorithm takes as input the graph $\mathcal{G} = (\mathcal{V}, \mathcal{E}, \mathbf{X})$, pre-computed propagated features $\tilde{\mathbf{X}}$ (as described in Section 4.2 via Eq. 6), and Global Influence Scores $\{\mathcal{I}_g(v_i)\}_{v_i \in \mathcal{V}}$ (computed via Eq. 5 in Section 4.1). The student MLP is trained using the supervised loss (Eq. 7) and distillation loss (Eq. 8), weighted by the hyperparameter $\lambda$ as specified in Eq. 9.

---

**Algorithm 1** InfGraND: Influence-Guided GNN-to-MLP Knowledge Distillation

---

1: **Input:** Graph $\mathcal{G} = (\mathcal{V}, \mathcal{E}, \mathbf{X})$, labels $\mathbf{Y}_{lab}$ for $\mathcal{V}_{lab}$, propagated features $\tilde{\mathbf{X}}$, influence scores $\{\mathcal{I}_g(v_i)\}_{v_i \in \mathcal{V}}$
2: **Input:** Hyperparameters: $\lambda, \gamma_1, \gamma_2, \delta_1, \delta_2, \tau, E_{max}, \eta$
3: **Output:** Trained student MLP $f_{student}$
4: Train teacher GNN $f_{teacher}$ on $\mathcal{G}$ until convergence; freeze parameters
5: Compute teacher logits: $\mathbf{h}_i^t \leftarrow f_{teacher}(\mathcal{G}, v_i)$ for all $v_i \in \mathcal{V}$
6: Initialize student MLP $f_{student}$ with parameters $\theta$
7: **for** epoch $e = 1$ to $E_{max}$ **do**
8:      Compute student logits: $\mathbf{h}_i^s \leftarrow f_{student}(\tilde{\mathbf{x}}_i)$ for all $v_i \in \mathcal{V}$
9:      $\mathcal{L}_s \leftarrow \delta_1 \sum_{v_i \in \mathcal{V}_{lab}} D_{CE}(\sigma(\mathbf{h}_i^s), y_i) + \delta_2 \sum_{v_i \in \mathcal{V}_{lab}} \mathcal{I}_g(v_i) \cdot D_{CE}(\sigma(\mathbf{h}_i^s), y_i)$      $\triangleright$ Eq. 7
10:      $\mathcal{L}_d \leftarrow \sum_{i \in \mathcal{V}} \sum_{j \in \mathcal{N}(v_i)} (\gamma_1 + \gamma_2 \cdot \mathcal{I}_g(v_j)) \cdot \frac{1}{|\mathcal{N}(v_i)|} \cdot D_{KL}(\sigma(\mathbf{h}_i^s/\tau) \| \sigma(\mathbf{h}_j^t/\tau))$      $\triangleright$ Eq. 8
11:      $\mathcal{L}_t \leftarrow \lambda \mathcal{L}_s + (1 - \lambda)\mathcal{L}_d$      $\triangleright$ Eq. 9
12:      $\theta \leftarrow \theta - \eta \nabla_\theta \mathcal{L}_t$
13: **end for**
14: **return** $f_{student}$

---

## H  Sensitivity Analysis of Influence Computation Depth

To evaluate the sensitivity of InfGraND to the influence computation depth $k$ (Section 4.1), we conduct ablation experiments varying $k$ from 1 to 4 on Cora and Pubmed datasets using GCN and GraphSAGE teachers. As shown in Figure 9, performance remains relatively stable across different $k$ values, with $k = 2$ achieving the best or near-best results in most

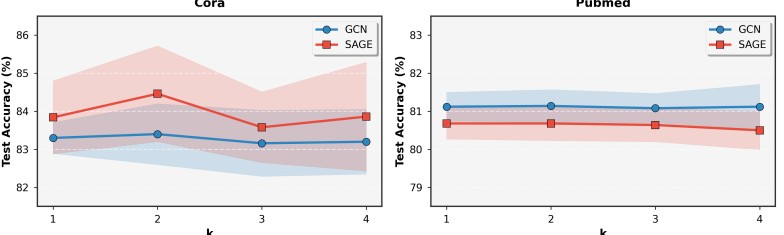

Figure 9: Ablation study on influence computation depth (k-hop) for Cora and Pubmed datasets using GCN and GraphSAGE teachers. Shaded regions represent standard deviation across multiple runs.

cases. The performance variation across $k \in [1, 4]$ is less than 1% for all configurations, confirming that our method is robust to this hyperparameter choice. These results validate our design decision to use $k = 2$ (Section 4.1),

