# OpenReview forum: "InfGraND: An Influence-Guided GNN-to-MLP Knowledge Distillation"
_TMLR — Accepted by TMLR_

### Review · Reviewer_JHV2 · 2025-09-15

**Summary Of Contributions:**

# Strengths

In general, this paper:

1. Keeps inference efficient by distilling to MLPs with preprocessed features, practical for deployment.

2. Includes good ablations and sensitivity analyses, showing thoughtful evaluation design.

**Audience:**

Yes

**Audience Explanation:**

This paper introduces a new angle by weighting knowledge transfer from GNN to MLP with node influence instead of just teacher uncertainty. Thus, there will be TMLR audiences interested in this paper.

**Broader Impact Concerns:**

Broader Impact Concerns

1. May bias performance toward well-connected nodes, leaving sparse nodes underrepresented.

2. Could be vulnerable to adversarial graph manipulations that inflate node influence.

3. No major ethical or misuse risks beyond typical concerns in graph learning.

**Claims And Evidence:**

Yes

**Claims Explanation:**

I believe this paper demonstrates consistent accuracy improvements, in some cases surpassing teacher GNNs.
And overall, the writing and paper structure are clear, with solid experimental coverage.

**Requested Changes:**

# Major Weaknesses

1. This work is dependent on homophily, not tested on heterophilous graphs. The method assumes neighbors share labels, so results may not generalize.

2. Heuristic influence metric. The influence is approximated via simplified propagation, which may not align with how GNNs actually weigh nodes. No direct validation.

3. Missing relevant baselines. Key structure-aware MLP methods like SA-MLP are absent, weakening comparative claims.

4. No handling of teacher errors. Weighting ignores teacher confidence, so mistakes may be reinforced.



# Minor Weaknesses

1. Incremental contribution: This work mainly integrates existing ideas (influence, propagation, KD) rather than introducing a fundamentally new technique.

2. Narrow task scope: Only node classification is evaluated, leaving generalizability to other tasks unclear.

3. No ablation against simpler metrics: Unclear if degree/PageRank would perform similarly.

4. Some clarity issues: Definitions are dense, and contributions could be presented more cleanly.

---

> ### Author Response · Authors · 2025-11-13
> **Rebuttal by Authors**
>
> We thank the reviewers for their thoughtful comments and constructive feedback. We have addressed each concern in detail below and highlighted the corresponding revisions in yellow within the manuscript.
>
> **Major W1:**
>
> **Response:** We thank the reviewer for raising this point. We acknowledge that our experiments focus on homophilic graph datasets, which is consistent with the standard benchmarks used in related work [1][2][3][4][5]. We have not claimed generalization to heterophilous graphs, and we have revised the abstract/introduction/conclusion to explicitly clarify this scope. Specifically, we will clearly state that our experimental evaluation focuses on homophilous datasets. We agree that evaluating on heterophilous graphs would be valuable. We plan to explore this in future work.
>
> **Major W2:**
>
> **Response:** We thank the reviewer for this comment. The design choice of our influence metric is grounded in both theory and practical considerations. The formal definition of node influence is the expected Jacobian between two nodes, a measure that is computationally intractable and requires an efficient approximation. Our goal in this approximation is to measure a node's intrinsic and task-agnostic structural importance, rather than its task-specific relevance. _An influence measure derived from the fully trained GNN teacher would be biased by the downstream task_. Relying on the GNN for both the distillation signal and the influence weighting would excessively emphasize the teacher's learned feature representations. Instead, we designed the influence metric to provide a complementary topology-based signal, ensuring that the student learns from both the teacher's task-specific knowledge and the graph's fundamental structure.
>
> Our design is well-supported by both established literature and direct empirical validation provided in paper. The influence measurement framework is inspired by prior work in influence maximization for active learning [6]. More critically, _as direct validation_, we demonstrate our metric's effectiveness in Section 5.2.1. The results, presented in Figure 2, show that GNNs trained exclusively on a subset of high-influence nodes consistently and significantly outperform models trained on low-influence nodes. This provides empirical evidence that our influence score successfully identifies the most critical nodes for GNNs generalization, directly motivating our distillation strategy.

---

> > ### Author Response · Authors · 2025-11-13
> > **Rebuttal by Authors**
> >
> > **Major W3:**
> >
> > **Response:** We discuss SA-MLP in our related work section. However, we were unable to include it as a baseline because of the lack of a publicly available official implementation, which is necessary for reliable reproduction. Furthermore, citing results directly from the original paper would violate our evaluation protocol and lead to an unfair comparison. To ensure scientific rigor, we re-ran all baselines from their public codebases to test them under identical experimental settings, data splits, and random seeds. For a valid knowledge distillation comparison, it is essential that all student models learn from the same teacher GNN, a condition that cannot be guaranteed when using previously published scores.
> >
> > Our comparative analysis focused on the most recent and relevant baselines for non-uniform distillation such as KRD and HGMD-mixup. While structure-aware MLPs are a related category, we benchmarked our method against NOSMOG as a representative of this class in Table bellow. Since the original NOSMOG study used a different set of datasets, we report its transductive results only for those datasets that overlap with our evaluation. The results in the table confirm that InfGraND outperforms the NOSMOG baseline.
> >
> >
> > ## Table 1: Node classification accuracy (%) for various models under transductive setting
> >
> > | **Teacher** | **Method**   | **Cora**       | **Citeseer**   | **Pubmed**     | **Amazon**     | **CS**         | **Phy**        | **Avg Δ** |
> > | ----------- | ------------ | -------------- | -------------- | -------------- | -------------- | -------------- | -------------- | --------- |
> > | **GCN**     | Vanilla GCN  | 82.2 ± 0.6     | 71.6 ± 0.2     | 79.2 ± 0.3     | 90.7 ± 0.3     | 89.3 ± 0.0     | 91.9 ± 1.3     | +3.0      |
> > |             | Vanilla MLP  | 57.8 ± 1.0     | 60.5 ± 0.7     | 72.8 ± 0.4     | 79.0 ± 1.0     | 87.8 ± 0.5     | 89.5 ± 2.0     | +12.6     |
> > |             | GLNN         | 83.1 ± 0.3     | 73.0 ± 0.5     | 79.4 ± 0.6     | 92.3 ± 0.5     | 92.6 ± 0.4     | 93.6 ± 1.1     | +1.5      |
> > |             | KRD          | 83.3 ± 0.9     | 73.9 ± 0.8     | 81.8 ± 0.4     | 91.7 ± 1.5     | 93.1 ± 0.5     | 94.1 ± 0.3     | +0.8      |
> > |             | FF-G2M       | 83.5 ± 0.7     | 74.0 ± 0.5     | 79.9 ± 0.4     | 93.0 ± 0.2     | 93.0 ± 0.5     | 93.7 ± 1.5     | +1.0      |
> > |             | HGMD-mixup   | 83.9 ± 2.0     | 74.6 ± 0.1     | 81.9 ± 0.2     | 92.3 ± 1.3     | 93.1 ± 0.5     | 93.4 ± 1.3     | +0.6      |
> > |             | NOSMOG       | 83.7 ± 0.3     | 73.1 ± 1.8     | 78.1 ± 2.8     | 93.7 ± 0.5     | --             | --             | +1.0      |
> > |             | **InfGraND** | **84.0 ± 0.5** | **75.2 ± 1.1** | **81.3 ± 0.2** | **94.2 ± 0.4** | **93.5 ± 0.6** | **94.7 ± 0.0** | --        |
> > | **SAGE**    | Vanilla SAGE | 82.5 ± 0.6     | 70.8 ± 0.6     | 77.9 ± 0.4     | 92.6 ± 0.3     | 89.7 ± 0.0     | 92.0 ± 0.9     | +2.8      |
> > |             | Vanilla MLP  | 57.8 ± 1.0     | 60.5 ± 0.7     | 72.8 ± 0.4     | 79.0 ± 1.0     | 87.8 ± 0.5     | 89.5 ± 2.0     | +12.5     |
> > |             | GLNN         | 83.2 ± 0.9     | 70.4 ± 1.9     | 79.2 ± 0.5     | 92.4 ± 0.5     | 92.3 ± 1.0     | 93.6 ± 1.5     | +1.9      |
> > |             | KRD          | 83.6 ± 1.0     | 73.8 ± 0.6     | 80.9 ± 0.5     | 91.7 ± 1.3     | 93.2 ± 0.7     | 94.1 ± 1.0     | +0.9      |
> > |             | FF-G2M       | 83.9 ± 0.8     | 72.8 ± 0.6     | 79.5 ± 0.5     | 92.3 ± 0.7     | 92.8 ± 0.7     | 93.5 ± 1.5     | +1.3      |
> > |             | NOSMOG       | 83.0 ± 1.2     | 73.4 ± 1.2     | 76.4 ± 3.8     | 92.9 ± 0.5     | --             | --             | +1.0      |
> > |             | **InfGraND** | **84.5 ± 0.6** | **74.3 ± 0.5** | **81.3 ± 0.4** | **94.6 ± 0.3** | **93.4 ± 0.5** | **94.5 ± 1.1** | --        |
> > | **GAT**     | Vanilla GAT  | 81.8 ± 1.2     | 70.4 ± 0.9     | 77.5 ± 0.2     | 87.6 ± 1.6     | 90.5 ± 0.0     | 91.9 ± 1.2     | +3.8      |
> > |             | Vanilla MLP  | 57.8 ± 1.0     | 60.5 ± 0.7     | 72.8 ± 0.4     | 79.0 ± 1.0     | 87.8 ± 0.5     | 89.5 ± 2.0     | +12.6     |
> > |             | GLNN         | 83.4 ± 0.4     | 70.6 ± 2.5     | 80.5 ± 2.4     | 91.5 ± 0.6     | 93.3 ± 0.5     | 93.3 ± 1.6     | +1.7      |
> > |             | KRD          | 83.0 ± 1.1     | 72.9 ± 0.6     | 81.4 ± 0.4     | 91.8 ± 1.4     | 94.3 ± 0.5     | 94.0 ± 1.3     | +0.9      |
> > |             | FF-G2M       | 83.5 ± 0.6     | 71.4 ± 1.4     | 80.9 ± 0.6     | 91.0 ± 0.6     | 93.0 ± 0.3     | 94.0 ± 1.5     | +1.5      |
> > |             | NOSMOG       | 83.4 ± 0.4     | **74.4 ± 2.2**     | 75.2 ± 3.0     | 93.0 ± 1.3     | --             | --             | +1.1      |
> > |             | **InfGraND** | **84.2 ± 0.5** | 73.9 ± 0.8 | **81.6 ± 0.5** | **94.5 ± 0.3** | **94.2 ± 0.6** | **94.4 ± 0.0** | --        |

---

> > > ### Author Response · Authors · 2025-11-13
> > > **Rebuttal by Authors**
> > >
> > > **Major W4:**
> > >
> > > **Response:** While our influence-guided weighting does not explicitly account for the teacher's prediction confidence, our empirical results demonstrate this is not a practical limitation. Our method consistently outperforms confidence-based approaches like KRD and HGMD across all datasets, indicating that structural importance provides a more reliable signal than prediction uncertainty, which can be miscalibrated. Our approach follows the standard assumption in knowledge distillation literature that teacher predictions are sufficiently reliable for effective knowledge transfer. Future work could explore a hybrid approach that discriminates based on both structural influence and prediction uncertainty, which we have included under future work in the conclusion.
> > >
> > > ---
> > >
> > > **Minor W1:**
> > >
> > > **Response:** The core novelty lies in fundamentally re-framing GNN-to-MLP distillation by introducing a new, graph-aware framework guided by structural influence. This is a clear departure from prior graph-agnostic methods that rely on prediction uncertainty. The novelty is not in the individual components, but in the design and validation of this framework itself, which includes a new influence-guided loss function (Eq. 8). We demonstrate empirically (Table 1 and 2) that this new paradigm is a non-trivial contribution, as our framework consistently outperforms existing state-of-the-art, uncertainty-based methods like KRD and HGMD.
> > >
> > > **Minor W2:**
> > >
> > > **Response:** We focused exclusively on node classification as it is the standard benchmark task used to evaluate methods in the GNN-to-MLP distillation literature [1][2][3][4][5][7]. This focus allows for the most direct and fair comparison against existing state-of-the-art approaches, ensuring our results are situated accurately within the context of prior work.

---

> > > > ### Author Response · Authors · 2025-11-13
> > > > **Rebuttal by Authors**
> > > >
> > > > **Minor W3:**
> > > >
> > > > **Response:** To test our proposed influence metric against simpler alternatives, we conducted a focused ablation study using normalized Degree and PageRank. To isolate the metric's impact, we disabled our feature propagation module (Section 4.2) and used only the influence-guided loss component of our framework. We then integrated the Degree and PageRank scores by using them in place of our influence score, $\mathcal{I}_g(v_i)$, within Equations 7 and 8. The comparative results for the transductive setting across all three teacher GNNs are presented in Table 8 (Appendix E).
> > > >
> > > > ### Table 8: Expanded Ablation Study on Influence Metrics (All Teachers, Transductive Setting)
> > > >
> > > > This table compares the performance of InfGraND using the proposed influence metric against simpler Degree and PageRank centrality metrics across all three teacher GNNs.
> > > >
> > > > | **Teacher** | **Dataset** | **KD w/Influence** | **KD w/Degree** | **KD w/PageRank** |
> > > > |--------------|--------------|--------------------|------------------|--------------------|
> > > > | **GCN** | Cora | **83.4 ± 0.8** | 81.1 ± 0.9 | 80.8 ± 0.7 |
> > > > |  | Citeseer | **74.6 ± 0.6** | 71.6 ± 0.3 | 70.7 ± 0.8 |
> > > > |  | Pubmed | **81.1 ± 0.4** | 79.7 ± 0.0 | 79.5 ± 0.2 |
> > > > | **GAT** | Cora | **84.0 ± 0.0** | 81.6 ± 0.7 | 79.7 ± 0.9 |
> > > > |  | Citeseer | **72.2 ± 1.9** | 69.3 ± 5.2 | 69.0 ± 5.0 |
> > > > |  | Pubmed | **80.6 ± 0.4** | 79.0 ± 0.6 | 79.8 ± 0.4 |
> > > > | **SAGE** | Cora | **84.4 ± 0.0** | 81.5 ± 1.3 | 81.1 ± 0.8 |
> > > > |  | Citeseer | **72.7 ± 1.1** | 70.2 ± 2.0 | 69.5 ± 2.5 |
> > > > |  | Pubmed | **80.3 ± 0.4** | 78.0 ± 1.7 | 79.9 ± 0.3 |
> > > >
> > > >
> > > > The results clearly demonstrate that our proposed influence metric consistently outperforms both Degree and PageRank across all datasets and teacher models. This validates that our metric captures a more effective signal for guiding knowledge distillation than these simpler, well-known centrality measures. We have added these results in the revised version of the paper in Appendix E.
> > > >
> > > >
> > > > **Minor W4:**
> > > >
> > > > **Response:**  We appreciate the reviewer's feedback. We have revised the introduction to present our contributions more clearly. The technical density reflects the precision required for our theoretical framework, and we believe this level of rigor is essential for ensuring soundness.
> > > >
> > > >
> > > > ---
> > > >
> > > >
> > > >
> > > > [1] Wu, L., Lin, H., Huang, Y., \& Li, S. Z. (2023, July). _Quantifying the knowledge in gnns for reliable distillation into mlps_. In International Conference on Machine Learning (pp. 37571-37581). PMLR.
> > > >
> > > > [2] Wu, L., Liu, Y., Huang, Y., Lin, H., Tan, C., \& Li, S. Z. _HGMD: Rethinking Hard Sample Distillation for GNN-to-MLP Knowledge Distillation_.
> > > >
> > > > [3] Wu, L., Lin, H., Huang, Y., Fan, T., \& Li, S. Z. (2023, June). _Extracting low-/high-frequency knowledge from graph neural networks and injecting it into mlps: An effective gnn-to-mlp distillation framework_. In Proceedings of the AAAI Conference on Artificial Intelligence (Vol. 37, No. 9, pp. 10351-10360).
> > > >
> > > > [4] Lu, W., Guan, Z., Zhao, W., \& Yang, Y. (2024, August). _Adagmlp: Adaboosting gnn-to-mlp knowledge distillation._ In Proceedings of the 30th ACM SIGKDD Conference on Knowledge Discovery and Data Mining (pp. 2060-2071).
> > > >
> > > > [5] Tian, Y., Zhang, C., Guo, Z., Zhang, X., \& Chawla, N. V. (2022). _Nosmog: Learning noise-robust and structure-aware mlps on graphs_. arXiv preprint arXiv:2208.10010.
> > > >
> > > > [6] Zhang, W., Wang, Y., You, Z., Cao, M., Huang, P., Shan, J., ... \& Cui, B. (2021). _Rim: Reliable influence-based active learning on graphs._ Advances in neural information processing systems, 34, 27978-27990.
> > > >
> > > > [7] Zhang, S., Liu, Y., Sun, Y., \& Shah, N. (2021). _Graph-less neural networks: Teaching old mlps new tricks via distillation._ arXiv preprint arXiv:2110.08727.

---

### Review · Reviewer_gGmb · 2025-10-07

**Summary Of Contributions:**

This paper proposes a new influence-guided distillation approach to distill from GNN to MLP for efficient deployment of the models. Specifically, the approach computes an importance weight for each node in the graph by aggregating its influence for other nodes (through gradient estimation), and then directly multiplies this weight to the distillation loss for each node. The authors have conducted fairly comprehensive experiments across different models and graph datasets to illustrate the effectiveness.

### Strengths

1. The proposed approach is well-motivated and novel
2. The experiments in this work are very comprehensive, covering multiple graph datasets and GNNs, and compare with many baselines.
3. The empirical results are good

### Weaknesses

1. The loss functions from Eq 7 to 9 are a bit complicated and involve several hyper parameters. This brings extra difficulty for application of the proposed approach to tune multiple hyperparameters. I do have a question on these: are results in Table 1 all from one set of hyperparameters? Or each configuration tunes the hyper parameters on their own?

**Audience:**

Yes

**Audience Explanation:**

This paper is about a new approach of distilling MLP from GNNs which is common practice, thus this work should be interesting for some TMLR audience.

**Claims And Evidence:**

Yes

**Claims Explanation:**

The submission presents comprehensive evidence and ablation studies to support the claims.

**Requested Changes:**

As mentioned in the weaknesses, I suggest the authors to clarify more on the hyperparameter tuning process and explicitly state the final used set of hyperparameters values in the main experiments to obtain the results.

---

> ### Author Response · Authors · 2025-11-13
> **Rebuttal by Authors**
>
> We gratefully acknowledge the reviewer’s helpful feedback. We have revised the manuscript in response and submitted the updated version with the changes highlighted in yellow.
>
> **Weakness #1**
>
> **Response:** Thank you for this important question. Each configuration tunes its hyperparameters independently using validation accuracy as the optimization criterion. Details of our hyperparameter search spaces and tuning methodology are provided in Appendix B and Section 5.1 (Implementation).
>
> **Design Rationale:** Each hyperparameter serves a distinct purpose. The hyperparameter $\lambda$ in Eq. 9 follows standard practice in the GNN-to-MLP distillation literature, balancing the supervised loss ($\mathcal{L}_s$) and distillation loss ($\mathcal{L}_d$) to regulate the training forces from both ground-truth labels and teacher predictions. For Eq. 7 and 8, our design reflects a core principle: we aim to perform importance-based distillation without completely ignoring any nodes.
>
> For example, in Eq. 8, we use two hyperparameters ($\gamma_1$ and $\gamma_2$) rather than one. The term $\gamma_1$ provides a baseline distillation gradient from all neighbors, ensuring that even low-influence nodes contribute to learning. The term $\gamma_2 \cdot \mathcal{I}_{g}(v_j)$ acts as an amplifier that provides additional emphasis to high-influence nodes.
>
> This dual-component design prevents the complete suppression of knowledge from low-influence nodes while still prioritizing structurally important ones. In Appendix D.2, we provide a detailed theoretical analysis showing how the influence score $\mathcal{I}_{g}(v_j)$ scales the gradient magnitude to amplify learning from influential neighbors.
>
> Furthermore, Appendix D.3 rigorously justifies the necessity of $\gamma_1$, demonstrating that when $\mathcal{I}_{g}(v_j) = 0$ (Eq. 27), the gradient signal from $\gamma_1$ remains available to drive training. The same design principle applies to $\delta_1$ and $\delta_2$ in Eq. 7 for the supervised loss.
>
> **Tuning Complexity:** While Eqs. (7–9) introduce several coefficients, they do not materially increase tuning complexity. Hyperparameter search is required in any case, and these additional terms are tuned in the same process. As shown in our sensitivity analysis (Appendix H, Fig. 6 and Table 3), model performance remains stable across reasonable parameter ranges.
>
>
> ---
>
> **Requested Changes**
>
> **Response:** We thank the reviewer for their comment regarding hyperparameter transparency. In response to your request, we have provided Table 6 in Appendix B, which reports the final values for the five core InfGraND hyperparameters ($\lambda$, $\delta_1$, $\delta_2$, $\gamma_1$, $\gamma_2$) across all experimental configurations. The complete set of hyperparameters required for full reproducibility will be released with our code upon publication.
>
> As detailed in Section 5.1 and Appendix B, we performed independent tuning for each teacher-dataset configuration using the WandB platform, with validation accuracy as our selection criterion. To ensure fair comparison, the same trained teacher model was used across all student methods for a given experiment.
>
>
> ## Table 6: Hyperparameters for InfGraND
>
> | Dataset  | Teacher | **λ** | **δ₁** | **δ₂** | **γ₁** | **γ₂** | **λ** | **δ₁** | **δ₂** | **γ₁** | **γ₂** |
> |-----------|----------|------|------|------|------|------|------|------|------|------|------|
> |           |          | **Transductive** | | | | | **Inductive** | | | | |
> | **Cora**  | GCN  | 0.1 | 0.6 | 0.2 | 0.8 | 0.4 | 0.1 | 1.0 | 0.2 | 0.8 | 0.3 |
> |           | SAGE | 0.5 | 0.8 | 0.1 | 1.0 | 0.4 | 0.5 | 1.0 | 0.2 | 1.0 | 0.2 |
> |           | GAT  | 0.5 | 0.4 | 0.2 | 1.0 | 0.1 | 0.5 | 0.9 | 0.8 | 1.0 | 0.2 |
> | **Citeseer** | GCN  | 0.0 | 0.6 | 0.1 | 0.6 | 0.4 | 0.0 | 1.0 | 0.2 | 0.8 | 0.3 |
> |           | GAT  | 0.1 | 0.8 | 0.2 | 0.6 | 0.2 | 0.1 | 0.9 | 0.8 | 1.0 | 0.2 |
> |           | SAGE | 0.1 | 0.6 | 0.1 | 0.4 | 0.4 | 0.0 | 1.0 | 0.2 | 1.0 | 0.2 |
> | **Pubmed** | GCN  | 0.0 | 1.0 | 0.1 | 0.6 | 0.4 | 0.5 | 1.0 | 0.2 | 0.2 | 1.0 |
> |           | GAT  | 0.0 | 0.4 | 0.2 | 0.4 | 0.1 | 0.0 | 0.5 | 0.2 | 0.2 | 1.0 |
> |           | SAGE | 0.0 | 0.8 | 0.1 | 0.8 | 0.2 | 0.0 | 0.1 | 0.1 | 0.8 | 0.1 |
> | **Amazon** | GCN  | 0.2 | 1.0 | 0.2 | 0.8 | 0.1 | 0.2 | 1.0 | 0.1 | 1.0 | 0.1 |
> |           | GAT  | 0.3 | 0.6 | 0.1 | 0.8 | 0.01 | 0.3 | 0.8 | 0.1 | 0.6 | 0.1 |
> |           | SAGE | 0.2 | 0.4 | 0.7 | 0.7 | 0.4 | 0.2 | 0.9 | 0.4 | 0.6 | 0.1 |
> | **CS**    | GCN  | 0.3 | 1.0 | 0.1 | 0.5 | 0.5 | 0.3 | 1.0 | 0.2 | 1.0 | 0.2 |
> |           | GAT  | 0.0 | 1.0 | 0.1 | 0.01 | 0.01 | 0.3 | 0.8 | 0.1 | 0.6 | 0.1 |
> |           | SAGE | 0.5 | 0.5 | 0.5 | 1.0 | 0.01 | 0.5 | 0.4 | 0.4 | 0.4 | 0.1 |
> | **Physics** | GCN  | 0.2 | 0.1 | 0.01 | 0.1 | 0.1 | 0.2 | 0.1 | 0.001 | 0.6 | 0.001 |
> |           | GAT  | 0.2 | 0.8 | 0.8 | 0.8 | 0.8 | 0.2 | 1.0 | 0.6 | 0.1 | 0.001 |
> |           | SAGE | 0.5 | 1.0 | 0.2 | 0.2 | 0.2 | 0.5 | 0.1 | 0.01 | 0.4 | 0.01 |

---

### Review · Reviewer_gJYB · 2025-11-02

**Summary Of Contributions:**

**Summary of Contributions**

This paper presents InfGraND, a novel framework for graph-aware knowledge distillation from Graph Neural Networks (GNNs) to Multi-Layer Perceptrons (MLPs). The core idea is to enhance the efficiency of MLP-based graph learning while retaining the representational strength of GNNs.

The authors introduce a node influence–guided distillation mechanism, which prioritizes structurally important nodes based on their topological impact within the graph rather than prediction uncertainty. This “influence score” captures how perturbations to one node affect others, leading to a more informed and stable distillation process.

In addition, InfGraND integrates a one-time precomputation of multi-hop neighborhood features, enabling the MLP student to leverage graph structure without runtime message passing. This design preserves low inference latency while embedding structural awareness.

The framework is evaluated comprehensively on seven benchmark datasets (including Cora, Citeseer, Pubmed, Amazon-Photo, Coauthor-CS/Phy, and OGBN-Arxiv) under both transductive and inductive settings, using standard GNN architectures (GCN, GAT, GraphSAGE) as teachers. Experimental results show that InfGraND consistently outperforms prior GNN-to-MLP KD methods and, in several cases, even surpasses the GNN teacher’s accuracy. Additional ablation and label-scarce experiments support the robustness of the approach.

**Key Strengths**

Novel contribution: Introduces a graph-structure–aware node influence metric for knowledge distillation — a conceptually sound and underexplored direction.

Balanced design: Maintains MLP’s low inference cost while meaningfully embedding structural knowledge.

Comprehensive evaluation: Covers multiple datasets, architectures, and training settings with clear experimental comparisons.

Clarity of motivation: The paper articulates the trade-off between GNN expressiveness and MLP efficiency convincingly.

**Key Weaknesses**

Limited theoretical analysis: The influence metric, though intuitive, lacks rigorous theoretical grounding (e.g., formal properties or complexity discussion).

Potential scalability concerns: Influence computation, even if parameter-free, may be costly for large-scale graphs; practical runtime analysis is not deeply explored.

Comparative novelty: Some components (feature precomputation, subgraph-level distillation) build upon prior ideas, so the degree of methodological novelty depends heavily on the influence-guided weighting.

**Audience:**

Yes

**Audience Explanation:**

this GNN related work is a mainstream research in machine learning

**Claims And Evidence:**

Yes

**Claims Explanation:**

**Overall assessment**:
Yes. The submission provides strong, well-structured empirical evidence supporting its major claims. The experiments are comprehensive, reproducible, and logically aligned with the stated goals. Evidence is clear and convincing across multiple datasets and evaluation settings.

**Claim 1 – The proposed influence metric identifies structurally important nodes.**

Evidence:

Q1 directly compares GNNs trained on high- vs. low-influence nodes.

High-influence subsets consistently yield higher accuracy across all datasets.
Reviewer analysis:
✓ Clear empirical validation; convincingly demonstrates that the influence score captures structural importance.

**Claim 2 – Influence-guided distillation improves MLP performance and can surpass GNN teachers.**

Evidence:

Q2 shows InfGraND consistently outperforms baselines (GLNN, KRD, HGMD, FF-G2M).

Distilled MLPs exceed teacher accuracy on several datasets.
Reviewer analysis:
✓ Strongly supported; results are comprehensive and cross multiple architectures and benchmarks.

**Claim 3 – InfGraND offers a superior accuracy–efficiency trade-off.**

Evidence:

Q3 reports 6.8×–13.9× faster inference with higher or comparable accuracy.

Evaluation uses consistent architecture and hyperparameter settings for fairness.
Reviewer analysis:
✓ Persuasive demonstration of low-latency advantage; efficiency claim is fully substantiated.

**Claim 4 – Both core components (influence weighting and feature propagation) contribute complementary gains.**

Evidence:

Q4 ablation study isolates each module: both yield improvements independently and achieve the best performance when combined.
Reviewer analysis:
✓ Well-designed ablation confirms contribution of each component; evidence is quantitative and coherent.

**Claim 5 – InfGraND remains robust under label scarcity.**

Evidence:

Q5 shows consistent 4% average gains over GLNN when only 2–8 labels per class are available.
Reviewer analysis:
✓ Strong indication of robustness; experiments are realistic and address an important real-world scenario.

**Claim 6 – InfGraND is stable across hyperparameter settings.**

Evidence:

Q6 hyperparameter analysis shows smooth trends for λ, γ₂, δ₂, and P; no abrupt degradation observed.

Optimal λ ≈ 0.1 and P = 2 generalize well.
Reviewer analysis:
✓ Supports robustness claim; results are interpretable and not overly sensitive to tuning.

**Final Reviewer Judgment**

All major claims are clearly supported by accurate and consistent experimental evidence.
The evaluation suite is broad, methodologically sound, and internally coherent.
Minor extensions—such as results on more diverse graph domains or runtime profiling of influence computation—could further strengthen the paper, but the presented evidence is convincing and sufficient to substantiate the authors’ conclusions.

**Requested Changes:**

**Proposed Adjustments: Critical (Essential for Acceptance)**

1. **Clarify computational cost of influence estimation.**

Provide explicit time/space complexity or runtime comparison for computing node influence, especially on large graphs such as OGBN-Arxiv.

From Section 4.1 (Node Influence Measurement):
The authors define the influence metric formally and then approximate it using cosine similarity between k-hop propagated features.
The authors say this approach is “efficient” and “parameter-free,” but no complexity expression or runtime profile is provided.
There is no mention of how long the influence computation takes on large graphs like OGBN-Arxiv, nor whether it scales linearly or quadratically in |V| or |E|.

In Section 5 (Experiments), no table or figure reports “influence computation time,” only inference time comparisons.

This is crucial to fully justify the “efficiency” claim and to ensure scalability is practical in real-world deployments.

2. **Add more details on the influence approximation procedure.**

Clarify exactly how cosine similarity–based influence is computed (e.g., hop size, normalization, batch implementation).

a. Exact hop count (k) used in cosine similarity is missing. The text mentions “for example, k = 2,” but it’s unclear whether this is fixed, tuned, or dataset-dependent.

b. For the normalization scheme, “MinMaxScaler” is mentioned, but over what scope? (per node, per graph, or per batch?) This affects reproducibility and scaling.

c. Batch computation / complexity: How is the cosine similarity matrix computed in practice — all-pairs, sampled neighbors, or matrix multiplication shortcut? The naive O(N²) version is infeasible for large graphs.

d. Handling of self-loops / adjacency normalization: It mentions “normalized adjacency” but not whether self-loops are added or what normalization scheme is used (row-, sym-, or degree-based).

e. Implementation pseudocode: No pseudocode or algorithm listing is given; only high-level equations. For a reproducibility-focused reader, these are helpful but not sufficient for exact replication.


3. **Improve clarity and conciseness of exposition.**

The introduction and related-work sections are overly dense and could better highlight the conceptual novelty (influence-guided vs. entropy-guided discrimination).


Streamlining these sections would enhance readability and strengthen the positioning of contributions.


**Non-Critical (Would Strengthen the Work)**

1. Extend evaluation to more graph domains.

Include at least one heterogeneous or dynamic graph dataset, or discuss limitations in generalizing beyond citation/co-author graphs.

This would broaden empirical validity.

2. Provide deeper theoretical insight.

A short analysis connecting the influence metric to established graph centrality or sensitivity measures would help ground the intuition mathematically.

3. Ablation on influence computation depth.

Currently, k-hop choice (k=2) is fixed; reporting how performance changes for k=1–4 in influence calculation would clarify stability.

---

> ### Author Response · Authors · 2025-11-13
> **Rebuttal by Authors**
>
> We sincerely thank the reviewer for the valuable comments and suggestions. We have revised the paper based on these comments and uploaded the updated manuscript with changes highlighted in yellow.
>
> **Proposed Adjustments: Critical #1: Clarify computational cost of influence estimation.**
>
> We thank the reviewer for this important observation. We provide explicit complexity analysis and comprehensive empirical validation below.
>
> **Complexity Analysis.** Our influence computation has two possible implementations: (a) the naive all-pair approach with complexity $O(N^2 \cdot D)$, which computes similarity between all node pairs and is feasible for small- and medium-sized graphs, (b) the k-hop approximation with complexity $O(|E_k| \cdot D)$, where $|E_k|$ denotes edges within k-hop neighborhoods. For sparse graphs, $|E_k| \ll N^2$, and for $k=2$, this is approximately $O(N \cdot \bar{d}^2 \cdot D)$ where $\bar{d}$ is the average degree. This k-hop approximation assumes that influence decays with distance, therefore, if we consider only close nodes, we would have most of a node's influence. Regarding space complexity, the naive approach requires $O(N^2)$ memory (~114 GB for OGBN-Arxiv with 169K nodes), while the k-hop approach requires only $O(|E_k|)$ (<1 GB for OGBN-Arxiv).
>
> _Influence computation is a one-time preprocessing step during training, not a recurring cost during inference_. Therefore, even if computation takes minutes or hours, it does not affect deployment latency or real-time prediction performance. This is why Section 5 focuses on inference time comparisons, as our efficiency claims pertain to deployment rather than training.
>
>
> **Empirical Validation.** We conducted comprehensive experiments across seven benchmark datasets using GPU-batched implementation on a single NVIDIA GPU (32GB VRAM) to report the computation time for different methods in different datasets. Table 9 reports the results. For OGBN-Arxiv (169K nodes, 2.5M edges), our 2-hop method completes is only 70 seconds (~1.2 minutes). The naive approach for OGBN-Arxiv is infeasible due to the 114 GB memory requirement. Across all datasets, our method scales from 2.7K to 169K nodes efficiently, validating its practical scalability. We have added a new appendix about the influence computation time, it is brought is Appendix F.
>
> **Design Choice for OGBN-Arxiv.** We use the 2-hop approximation for OGBN-Arxiv for three reasons: (1) the naive approach exceeds available GPU memory (114 GB required), (2) the 2-hop neighborhood captures most of influential interactions in homophilous graphs, (3) compared to stochastic sampling approaches, our 2-hop method is deterministic and reproducible.
>
> **Table 9:** Influence computation runtime (GPU-batched, single 32GB GPU). Times in seconds. *Naive approach for OGBN-Arxiv requires ~114 GB memory (infeasible).
>
> | Dataset | Nodes | Edges | Naive (s) | 1-hop (s) | 2-hop (s) | 3-hop (s) |
> |---------|-------|-------|-----------|-----------|-----------|-----------|
> | Cora | 2,708 | 13,264 | 5.68 | 0.95 | 1.18 | 2.34 |
> | Citeseer | 3,327 | 12,431 | 0.13 | 1.29 | 1.60 | 2.11 |
> | Pubmed | 19,717 | 108,365 | 5.25 | 9.69 | 13.48 | 34.06 |
> | Amazon-Photo | 7,650 | 245,812 | 0.89 | 3.75 | 22.93 | 202.60 |
> | Coauthor-CS | 18,333 | 182,121 | 5.39 | 9.42 | 14.70 | 63.28 |
> | Coauthor-Phy | 34,493 | 530,417 | 21.95 | 24.16 | 47.82 | 354.43 |
> | OGBN-Arxiv | 169,343 | 2,501,829 | N/A* | 49.78 | 70.13 | 498.46 |
>
> ---
>
>
>
> **Proposed Adjustments: Critical #2: Add more details on the influence approximation procedure.**
>
> Thank you for your valuable suggestions. We address each point below:
>
> **a:** We used a fixed $k=2$ hops for the influence calculation (Eq.4) across all datasets. This was a well-justified design, not a tuned hyperparameter, based on the observation, 2-layer GNNs are usually enough to get the optimal performance for these benchmarks, suggesting the 2-hop neighborhood contains the most critical feature interactions. We have revised the paper and clarified this in section 4.1.
>
>
> **b:** We have revised Section 4.1 for clarity. The normalization is applied globally per graph as a one-time offline step.
>
> **c:** As mentioned before in the response, for small/medium datasets (Cora, Citeseer, Pubmed, etc.), we use naive all-pairs cosine similarity computation ($O(N^2 \cdot D)$), which is feasible given their size. For OGBN-Arxiv, the naive approach is infeasible, so we use k-hop neighborhood approximation. This reduces computation from $O(N^2 \cdot D)$ to $O(|E_k| \cdot D)$. See Table 9 (from Appendix F) in our response above for runtime comparisons. We have added a new appendix about the influence computation, it is brought is Appendix F.

---

> > ### Author Response · Authors · 2025-11-13
> > **Rebuttal by Authors**
> >
> > **d**: The normalized adjacency is aligned with the common practice in GNNs. We revised the paper and added the defintion in the paper in section 4.1. We use the standard symmetric normalization with self-loops, following the approach from GCNs and SGC, where the normalized adjacency matrix $\tilde{A}$ is computed as $\tilde{A} = \tilde{D}^{-\frac{1}{2}}(A + I)\tilde{D}^{-\frac{1}{2}}$. We have added this change in Section 4.1. for clarity.
> >
> > **e**: Thank you for this valuable feedback. To improve reproducibility, we have added detailed pseudocode (Algorithm 1) in Appendix G. Furthermore, to ensure full reproducibility, we confirm that the code will be made public upon publication.
> >
> > ---
> >
> > **Proposed Adjustments: Critical #3: Improve clarity and conciseness of exposition.**
> >
> > Thank you for your comments. We have revised it accordingly. The changes are highlighted in yellow in the updated manuscript.
> >
> > ---
> >
> > **Proposed Adjustments: Non-Critical #1: Extend evaluation to more graph domains.**
> >
> > We appreciate your suggestion. We will pursue applying to other domains and graphs in future works. We have added this direction in the future works section.
> >
> > **Proposed Adjustments: Non-Critical #2: Provide deeper theoretical insight.**
> >
> > Thank you for this suggestion. We have theoretical analysis in Appendix D that provides mathematical grounding for our influence metric, including gradient derivation (D.1), analysis of how influence modulates gradient magnitude (D.2), and justification of the loss formulation (D.3).
> >
> > Regarding centrality measures: our influence metric differs fundamentally from traditional measures like Degree and PageRank, which are purely link-based and ignore node features. Our metric incorporates both features and topology through propagation. We empirically validate this distinction in Appendix E. We plan to explore deeper theoretical connections to sensitivity measures and centrality in future work.
> >
> > **Proposed Adjustments: Non-Critical #3: Ablation on influence computation depth.**
> >
> > Thank you for bringing this suggestion to our attention. We have conducted ablation studies on the influence computation depth ($k$=1-4) for Cora and Pubmed datasets with GCN and GraphSAGE teachers. The results show stable performance across different $k$ values, with $k$=2 achieving slightly better. These findings support our choice of $k=2$ as a reasonable default while demonstrating that InfGraND is robust to this hyperparameter. We have added these results in Appendix H (Figure 9).

---

### Decision · Action_Editor_NpYG · 2025-12-11

**Recommendation:** Accept as is

**Audience:**

Yes

**Audience Explanation:**

All Reviewers including myself think that there are individuals among the TMLR audience that will be interested in reading and knowing about the results of this paper.

**Claims And Evidence:**

Yes

**Claims Explanation:**

The authors provide experimental results to support their claims, and these were strengthened in their revision and responses. Their is also reflected by Reviewers' opinions and feedback.